# Multistability, intermittency, and hybrid transitions in social contagion models on hypergraphs

Guilherme Ferraz de Arruda ◎[1] ✉, Giovanni Petri ◎[1,2], Pablo Martin Rodriguez ◎[3] & Yamir Moreno ◎[1,4,5]

Although ubiquitous, interactions in groups of individuals are not yet thoroughly studied. Frequently, single groups are modeled as critical-mass dynamics, which is a widespread concept used not only by academics but also by politicians and the media. However, less explored questions are how a collection of groups will behave and how their intersection might change the dynamics. Here, we formulate this process as binary-state dynamics on hypergraphs. We showed that our model has a rich behavior beyond discontinuous transitions. Notably, we have multistability and intermittency. We demonstrated that this phenomenology could be associated with community structures, where we might have multistability or intermittency by controlling the number or size of bridges between communities. Furthermore, we provided evidence that the observed transitions are hybrid. Our findings open new paths for research, ranging from physics, on the formal calculation of quantities of interest, to social sciences, where new experiments can be designed.

How individuals interact in groups has motivated research in many different areas ranging from sociology and political sciences[1–5] to physics and mathematics[6–20]. From a sociological viewpoint, the interest frequently lies in the role played by committed minorities. One of the main questions is when and how this committed group of individuals can overturn a given consensus. Implicitly, we are assuming that the interaction between groups of individuals follows a critical-mass dynamics. Despite the informal use of the term critical-mass by politicians, the media, and even academics, there is evidence that individuals might behave in this way when changing social conventions. This evidence ranges from theoretical models[6–9] and observational studies[1,3,4,21,22], to real experimental approaches[5]. Although these studies suggest that the critical-mass threshold might range between 10% and 40%, there is evidence that it can be low as 0.3% in linguistic norm changes in English and Spanish[23,24] or even just a few of individuals that are not comparable with the size of the population under study[24,25]. Despite this wide range of observed thresholds, the critical-mass paradigm provides a reasonable abstraction to analyze and understand real social systems. Thus, from an analytical approach, we begin with the premise that the critical-mass dynamics is a reasonable assumption about how a group of people acts. So, the natural questions that emerge are: (1) How will a collection of groups behave? (2) How might the intersection between these groups change the global dynamics? (3) Can smaller groups have a higher critical-mass threshold than the whole population? Note that, as we allow for a collection of critical-mass dynamics, their intersections might be able to generate a cascade of events. In other words, by inducing change at a small scale, it might be possible to reach the threshold of other groups, therefore triggering global changes.

Recently, some of us proposed a formal model able to provide insights about these questions[12]. In this model, society is modeled as a hypergraph, where individuals are nodes, and the group interactions

[1]CENTAI Institute, Turin, Italy. [2]IMT Lucca, Lucca, Italy. [3]Department of Statistics, Federal University of Pernambuco (UFPE), Recife, PE, Brazil. [4]Institute for Biocomputation and Physics of Complex Systems (BIFI), University of Zaragoza, 50018 Zaragoza, Spain. [5]Department of Theoretical Physics, University of Zaragoza, 50018 Zaragoza, Spain. ✉e-mail: gui.f.arruda@gmail.com

of arbitrary sizes are encoded as hyperedges. The model presents discontinuous transitions, bistability, and hysteresis, thus, suggesting that interactions in groups might be the driver for such phenomenology, hence, already providing some initial answers and insights about question (1). In practice, the model suggests that some intermediate levels of activation are not reachable as the activation of groups might be able to trigger a larger scale cascade. Regarding questions (2) and (3), the model provides a theoretical foundation for, and a phenomenological explanation to, the seemingly different experimental findings of expected critical-mass thresholds. More specifically, ethnographic studies show a critical mass around 25–30%[1] and align with experimental results, which report a critical mass around 25%[5]. On the other hand, considering linguistic norm changes, the observed threshold is as low as 0.3% in English and Spanish[23]. The first studies consider a single group, while the linguistic norm changes consider a whole population, which can be understood as a collection of groups. Thus, in the latter, we might have groups with different sizes, each with a different threshold. In other words, from the perspective of the model in ref. [12], it is possible to have individual groups with thresholds between 25% and 40%, and, at the same time, due to the group intersections, having a critical mass at the population level around a much lower value. A second possible explanation is bi-stability, which enables two possible solutions for the same set of parameters. For example, the system might be operating in a region where both solutions are larger than zero and stable.

Here, through a dynamical analysis of the social contagion model presented in ref. [12], we show that the richness of this model is not constrained to discontinuous transitions and hysteresis. First, by evaluating a real hypergraph, we show that social contagion in hypergraphs can display a bimodal distribution of the number of active nodes, leading to multistability or intermittency in time. We also observe that at the transitions between branches, the susceptibility diverges. In the rest of the paper, we dedicate our efforts to give theoretical support to these findings and explain the mechanisms that might trigger them. We demonstrate that these features could be linked to the community structure in the hypergraph and we show that bridges between communities play a crucial role. Here, we define bridges as hyperedges that are composed by nodes belonging to different communities. Our second main result concerns the nature of the observed transitions. As we have multiple stable branches, due to the mentioned multistability, we might also have multiple transitions. Despite the expected discontinuities (see refs. [10,12–14]), we show that these transitions display features of hybrid transitions, that is, they display discontinuities and scaling behaviors for the order parameter and susceptibility.

The paper is organized as follows: in section "Model definition and theoretical analysis," we discuss the theoretical basis of the model presented in ref. [12], including its analytical and numerical aspects. In section "Example of real-world hypergraphs: the case of blues reviews," we present the numerical simulations we performed on a real hypergraph, which show evidence of multistability, intermittent behavior, and hybrid phase transitions. In the following sections, we focus on explaining our findings. In section "Multistability and intermittent behavior," we show that our first-order approximation predicts multistability. Next, using an artificial model, we relate multiple stable branches and intermittency to community structures. We also show how bridge hyperedges modulate the transition from multistability to intermittency. In section "Analysis of the transition between stable branches," we focus on a hypergraph with special symmetries, which allow us to derive exact equations for the dynamics and perform a finite size analysis, providing a strong argument for the presence of hybrid phase transitions in our model. Finally, in section "Discussion," we discuss our findings in more general terms, provide the conclusion, and show some of the perspectives opened by our work.

## Results

### Model definition and theoretical analysis

A hypergraph, $\mathcal{H}$, is defined as a set of nodes, $\mathcal{V} = \{v_i\}$ and a set of hyperedges $\mathcal{E} = \{e_j\}$, where $e_j$ is a subset of $\mathcal{V}$ with arbitrary cardinality $|e_j|$. The number of nodes is defined as $N = |\mathcal{V}|$. It is also convenient to define $\mathcal{E}_i$ as the set of hyperedges that contain the node $v_i$. If $\max(|e_j|) = 2$ we recover a graph. If for each hyperedge with $|e_j| > 2$ its subsets are also contained in $\mathcal{E}$, we recover a simplicial complex. Figure 1a shows an example of a hypergraph. Moreover, the adjacency matrix[13,26] can be defined as

$$\mathbf{A}_{ik} = \sum_{j\,:\,e_j \in \mathcal{E}_i \cap \mathcal{E}_k} \frac{1}{|e_j| - 1}, \tag{1}$$

for $i \neq k$, and $\mathbf{A}_{ii} = 0$ for all $i$. Note that it can be interpreted as a weighted projected graph. Here we will adopt this matrix for visualization purposes, but it has previously been used to study the spectra of hypergraphs[26] and linked to the stability of dynamical processes[13].

Our dynamics are defined through the activation and deactivation of nodes. In an arbitrary hypergraph, we associate a Bernoulli random variable $Y_i$ to each individual $v_i$ indicating whether the node $v_i$ is active ($Y_i = 1$) or not ($Y_i = 0$). For each active node, we associate a deactivation mechanism, modeled as a Poisson process with parameter $\delta$, $N_i^\delta$. For each hyperedge, $e_j$, we define a random variable $T_j = \sum_{k:v_k \in e_j} Y_k$, which is the number of active nodes in the hyperedge. See the tables next to each hyperedge in Fig. 1a for a graphical representation of all the possible microstates and the $T_j$ variables. If $T_j$ is equal or above a given threshold, $\Theta_j$, we associate a Poisson process with parameter $\lambda_j$, $N_j^{\lambda_j}$. We point out that the random variables defined above depend on $t$, for any $t \geq 0$, but we remove $t$ from our notation for the sake of simplicity. Formally, our dynamics can be written as a continuous-time Markov chain $(Y_t)_{t \geq 0}$, with state space $\{0,1\}^\mathcal{V}$. That is, for any $t \geq 0$, $Y_t$ is a random function from $\mathcal{V}$ into $\{0, 1\}$, which associates to each node $v_i$ the Bernoulli random variable $Y_i$. Moreover, the states of nodes change according to the following transitions and rates:

| Current state : | Transition : | Rate : |
|---|---|---|
| each active $v_i$ in $\mathcal{V}$ | $1 \to 0$ | $\delta$ |
| all inactive $v_k$ in $e_j$ | $0 \to 1$ | $\lambda_j \mathbb{1}_{\{T_j \geq \Theta_j\}}$ |

where $\mathbb{1}_{\{condition\}}$ is the indicator function, which is one if the "condition" is satisfied, and zero otherwise.

In other words, the group dynamics is given by a threshold process that becomes active only above a critical mass of activated nodes. When above the threshold, for a given hyperedge $e_j$, the Poisson process $N_j^{\lambda_j}$ induces that all the nodes inside this hyperedge become activate. So, given an hyperedge $e_j$, after the threshold is hit ($T_j \geq \Theta_j$) and a random time exponentially distributed with parameter $\lambda_j$ passes (as a consequence of the Poisson process $N_j^{\lambda_j}$), all the the inactive vertices become activate simultaneously. If enough nodes are deactivated before the time associated with the process passes, the process is removed. Moreover, if $|e_j| = 2$, we consider that the Poisson processes are directed. This definition allows for recovering traditional SIS contagion models. Figure 1b shows an example of the graphical representation for our process.

For simplicity, we assume that $\lambda_j = f(|e_j|)$, where $f$ is an arbitrary function of the cardinality of the hyperedge. It is also convenient to define $\Theta_j = \lceil \Theta^* |e_j| \rceil$, where $\lceil x \rceil$ is the ceiling function, which returns the least integer greater than or equal to $x$ and $\Theta^*$ is a global parameter that is invariant to the cardinality of the hyperedges and lies in the range $0 \leq \Theta^* \leq 1$. Note that, if we had defined $\Theta_j = |e_j| - 1$ we would have recovered a model similar to the one proposed in ref. [10]. It would be the same if $\mathcal{H}$ is constrained to a simplicial complex. For more on this relationship,

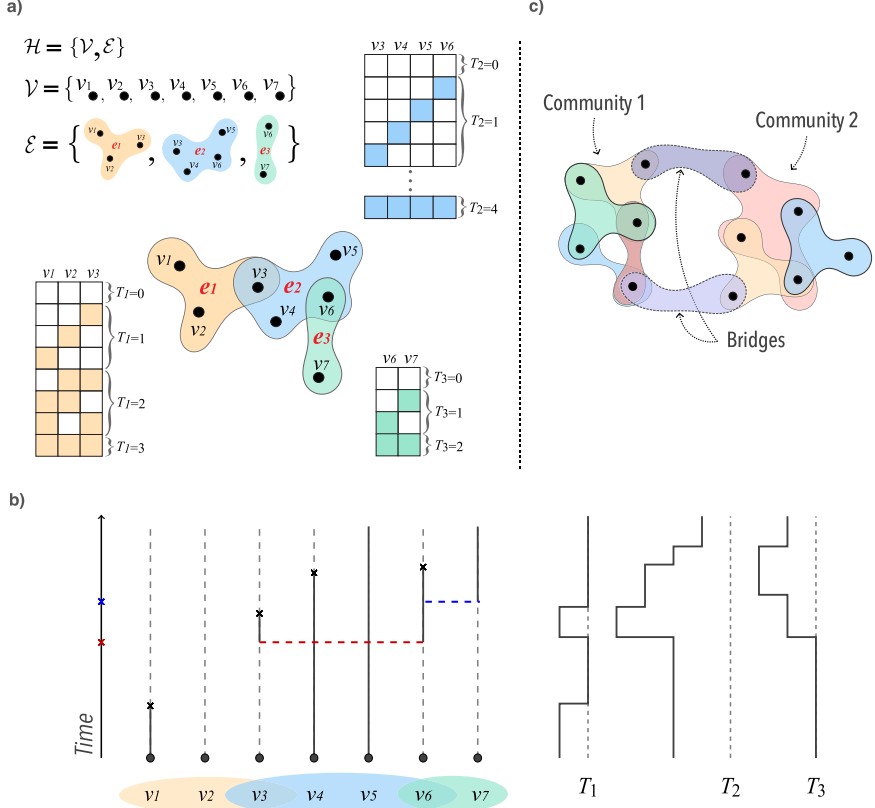

**Fig. 1 | Graphical example of the social contagion model on a hypergraph.** In **a**, we present the example of a hypergraph. The tables next to each hyperedge and with the same color represent all the possible microstate configurations and its respectively associated group variable $T_j$. In **b**, we show the graphical representation of one exemplary instance. In this representation, the black crosses represent the deactivation processes, $N_i^\delta, 1 \to 0$. For node $i$, the dashed lines represent inactive nodes, $Y_i = 0$, while continuous lines represent active nodes, $Y_i = 1$. In this specific example, the critical-mass threshold is $\Theta^* = 0.5$, the initial conditions are

$Y_1 = Y_4 = Y_5 = 1$ and $Y_2 = Y_3 = Y_6 = Y_7 = 0$, and the red and blue crosses mark the time that the processes $N_j^{\lambda_j}$ activate all the inactive nodes in $e_2$ and $e_3$, respectively. Moreover, on the right side of **b**, we show the temporal evolution of the $T_j$ variables in our exemplary instance. In this case, dashed lines indicate when the $T_j$s are equal to zero, each movement to the left represents an increase in $T_j$, while each movement to the right indicates a decrease. In **c**, we show a graphical example of the concept of bridges for two communities. Bridges are hyperedges that connect two communities, or groups of densely connected nodes.

we refer to refs. [12,14]. The exact equation that describes our model can be formally written as

$$\frac{d\langle Y_i \rangle}{dt} = \left\langle -\delta Y_i + (1 - Y_i) \sum_{j:e_j \in \mathcal{E}_i} \lambda_j \sum_{B_j} \mathbb{1}_{\{Y_i = 0, T_j \geq \Theta_j\}} \right\rangle, \quad (2)$$

where the first summation is over all hyperedges containing $v_i$ and the second summation is over all the possible dynamical micro-states inside the hyperedge $e_j$, denoted by the set $B_j$. Furthermore, $\mathbb{1}_{\{Y_i = 0, T_j \geq \Theta_j\}}$ is an indicator function that is 1 if $Y_i = 0$ and the critical-mass in the hyperedge is reached, and 0 otherwise. Moreover, we assumed that the spreading rate is composed by the product of a free parameter and a function of the cardinality, i.e., $\lambda_j = \lambda \times \lambda^*(|e_j|)$. In all of our numerical simulations we assumed $\lambda^*(|e_j|) = \log_2(|e_j|)$. This definition is convenient as, in the pairwise case, $\lambda^*(2) = 1$, guaranteeing that our dynamics reduces to the standard SIS model in a graph. Also, we choose $\log_2(|e_j|)$ as it grows sublinearly. So, in the limit of a large hyperedge, the average spreading rate tends to zero, i.e., $\lim_{|e_j| \to \infty} \frac{\log_2(|e_j|)}{|e_j|} = 0$.

### Example of real-world hypergraphs: the case of blues reviews
In this section, we present evidence that the behavior of real hypergraphs goes beyond the already surprising discontinuous transitions and bistability found in hypergraph and simplicial contagion models[10,12]. Indeed, we found that in many regimes our model presents

multiple stable solutions and regions of intermittent behavior, where we have an alternating dynamics of high and low activity. We divide our study in two parts: we begin with a macrostate analysis, and then move to a micro-state evaluation. This approach allows us to formulate some hypotheses regarding the mechanisms behind the observed phenomenology.

We first present evidence of multistability and intermittency in a real system. We do this by analyzing the dynamics of our model on the blues reviews hypergraph, where nodes are Amazon reviewers, and hyperedges are groups of reviewers who reviewed a certain type of blues music within a month[27]. This dataset is available at ref. [28]. This hypergraph has $N = 1106$ nodes and 694 hyperedges, whose maximum cardinality is $\max(|e_j|) = 83$. In this dataset, the pairwise interactions are sparse, which alone would form a giant component of only 24 nodes. However, by accounting all the hyperedges, the giant component of the hypergraph has $N = 1106$ nodes. We remark that repeated hyperedges were not allowed. Moreover, for a structural analysis of this hypergraph, we refer to the Supplementary Information (SI).

Figure 2 shows the QS Monte Carlo simulations (see the "Methods" section for more details about this method) for our social contagion model in the blues reviews hypergraph and in a randomly rewired version obtained from the exact version of the vertex-labeled hypergraph configuration model presented in ref. [29] (Algorithm 2 in ref. [29] and code from ref. [30]) after $10^7$ rewirings. Moreover, in the SI we present 30 additional Monte Carlo simulations for different randomizations of the blues reviews hypergraph, showing that they have a

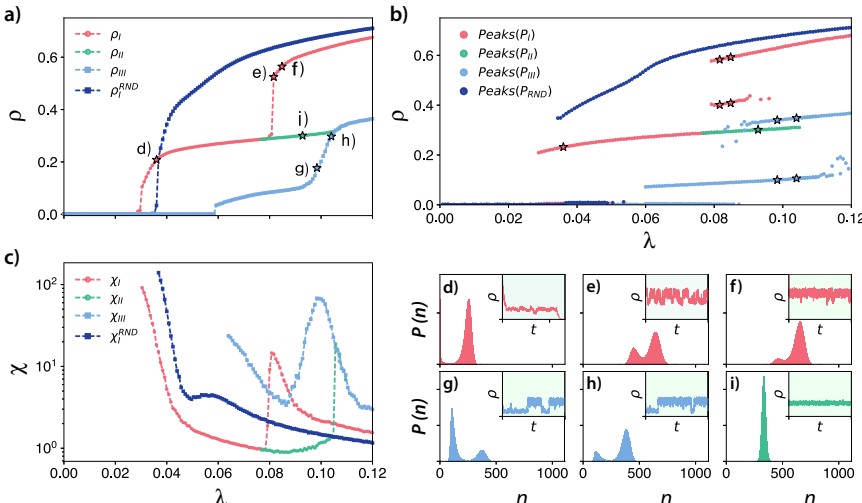

**Fig. 2 | Monte Carlo simulations for the social contagion model in the blues reviews hypergraph showing multistability and intermittency.** In **a**–**c**, we present the order parameter, the peaks of the state distributions, and the susceptibility, respectively. In **d**–**i**, we show the state distributions marked in **a**. In the insets of these plots, we show a small example of temporal behavior that generate these distributions. We performed the simulations using the real blues review hypergraph and a random version of it using $\delta = 1.0$, $\Theta^* = 0.5$, and $\lambda^*(|e_j|) = \log_2(|e_j|)$.

similar behavior. In a–c, we present the order parameter (average fraction of active individuals), peaks of the state distributions, and the susceptibility, respectively (see the "Methods" section for the formal definition of these quantities). In the remaining subplots, we present examples of state distributions for the points marked in (a). Observe that only one solution was found in the randomized version of the hypergraph, and it has a single discontinuous transition. This behavior contrasts with the real case, where multiple stable solutions, and multiple transitions between branches, were found. The comparison between the real case and the rewired version suggests that correlations play a significant role in the emergence of multistability and intermittency.

Considering the real scenario, we notice that our process recurrently presents a bimodal distribution of states, where the probability of the modes change as we increase or decrease $\lambda$. For Branch I, light red curves in Fig. 2, we notice that by increasing $\lambda$, we have a discontinuous transition. The distribution of states for this region is shown in Fig. 2d, where $\lambda = 0.036$ is used as an example. This distribution is bimodal, where the first mode is near the absorbing state (near $n \approx 1$ as we are using the QS method), and the second mode has an average $n \approx 250$. For lower values of $\lambda$, a similar picture is observed, but the probability of the first mode is higher than for the second. For higher values of $\lambda$, the opposite happens. This pattern is reproduced until we have a single mode with a bell-shaped distribution, similar to Fig. 2i. Next, as we increase $\lambda$, depending on the initial conditions and stochastic fluctuations the order parameter can jump (Fig. 2e, f). In this case, we again observe a bimodal distribution. In Fig. 2e, f, we show the state distribution for $\lambda = 0.0816$ and $\lambda = 0.0848$, respectively. Note that the mechanism that causes the bimodality in (d) is different from (e) and (f). In the first case, the bimodality appears as a consequence of the absorbing state, and it is similar to what is observed in an SIS process in a network. Note that for an SIS process in a network the second mode would be closer to the absorbing state and would increase continuously, originating a second-order phase transition. In the second case, the different modes are related to intermittent behavior, where the process oscillates between high and low activity regimes, as can be seen in the insets of these figures.

We also observed similar intermittent behavior in Branch III (Fig. 2a, g, h). Although branches I and III display intermittency, in the first branch, this implies a discontinuity in the susceptibility, while in the second, it generates a continuous peak of susceptibility, as can be observed in Fig. 2c. This peak of susceptibility is related to the time the system spends in the high or low activity regimes. In other words, the relative time the system spends in each mode changes the variance and, therefore, the susceptibility. Note that, in an SIS process in networks, similar susceptibility bumps are related to localization features of the network. For instance, in a network with communities, we could find a similar pattern. In this case, the bumps would suggest that the process manages to reach a community or a group of nodes[31]. Here, we use the term localization to denote a state where most of the probability of activation may be found within a constrained region, i.e., a subset of nodes. Note that, in graphs, we are usually interested in the localization properties at the transition, which can be quantified by the inverse participation ratio[32]. In our case, for simplicity, we are extending the word localization to characterize the supercritical regime.

We remark that the absorbing state is always accessible. For the initial condition $\rho = 0.1$, all the simulations fall in the absorbing state. This solution was not presented in Fig. 2 because it is trivial, and the susceptibility is noisy, possibly confusing the interpretation.

As we have intermittency and bimodal distributions, the order parameter alone might not be enough to fully describe our dynamical behavior. To better understand the behavior of our model, we also show the position of the peaks of the state distributions. These peaks represent the states in which the system is "locally more likely to be". For the blues reviews hypergraph, these peaks are reported in Fig. 2b. Despite the importance of the peaks, we argue that $\rho$ is still an essential global measurement for our dynamics. The order parameter, $\rho$, unambiguously defines the state of our system, while the same cannot be said about $Peaks(P)$. Notice that, in the multistable regions, the dynamics is not able to stay indefinitely in a single curve in Fig. 2b, as, in this case, the state is jumping between different modes. So, we argue that $\rho$ and $Peaks(P)$ should always be presented together.

Figure 2b shows that the bimodal distribution is present for a wide range of parameters, presenting regions where they vary continuously and regions with jumps. Moreover, in some cases, the different branches in Fig. 2b might be close to each other but can only be obtained in different simulations (see $Peaks(P_{II})$ and $Peaks(P_{III})$ in Fig. 2b). This observation suggests that the dynamics might be localized in different sets of nodes in the hypergraph. In this way, we might have similar macro-states as a consequence of significantly different micro-states.

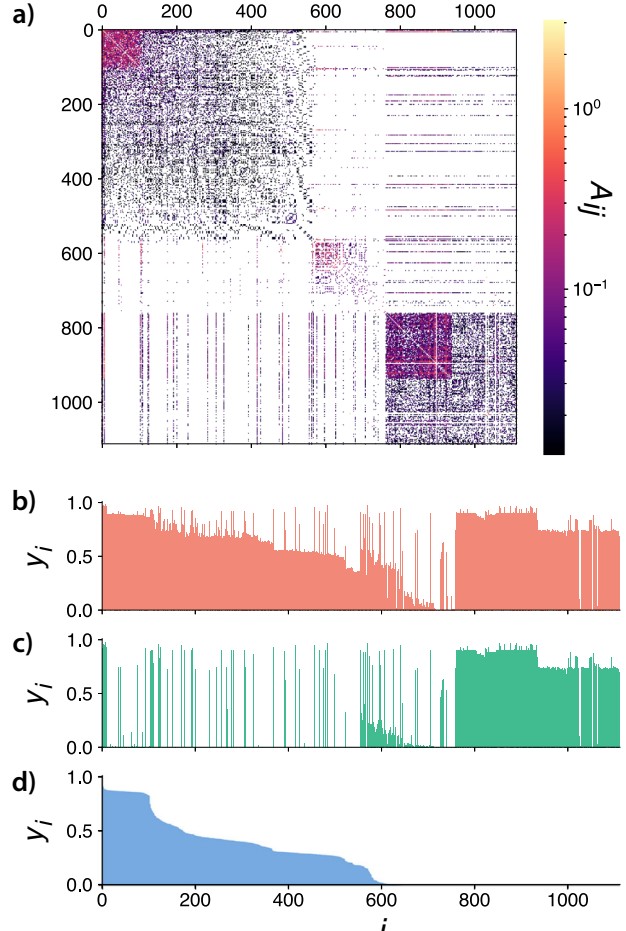

**Fig. 3 | Structure and dynamics of the blues reviews hypergraph.** In **a**, we plot the adjacency matrix of the blues reviews hypergraph ordered according to the activity of Branch III. In **b**–**d**, we show the probability of being active in branches I to III respectively. The probabilities were estimated using Monte Carlo simulations with $\lambda = 0.1016$, $\delta = 1.0$, $\lambda^* = \log_2(|e_j|)$ and $\Theta^* = 0.5$.

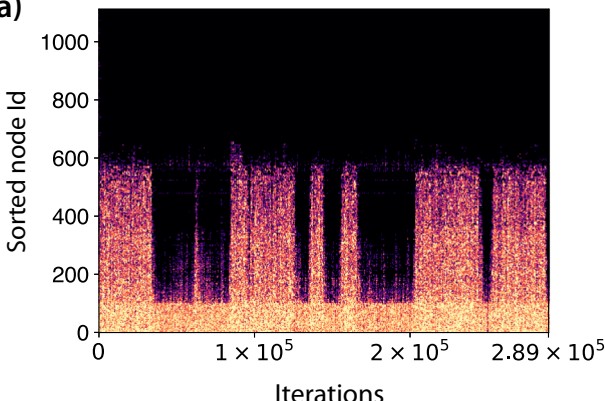

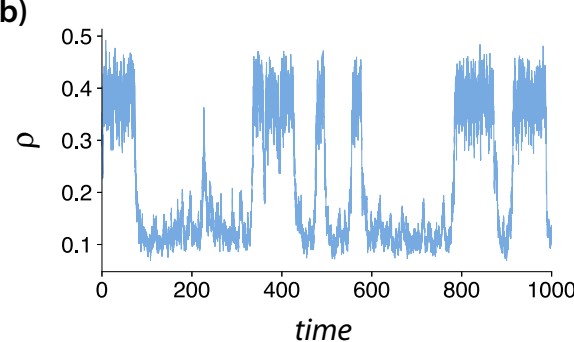

**Fig. 4 | Intermittent behavior in the blues reviews hypergraph.** In **a**, we show the graphical visualization of the activity of the nodes in a specific run of the MC. Pixels in dark represent inactive nodes, while colored pixels represent active nodes. The intermediate colors in the figure are only an effect of aliasing and do not have a physical interpretation. The nodes are sorted by their activity for visualization purposes. In **b**, we represent the order parameter as a function of time for the same simulation. The dynamical parameters for this simulation are $\lambda = 0.1016$, $\delta = 1.0$, $\Theta^* = 0.5$, and $\lambda^*(|e|) = \log_2(|e_j|)$. Notice that, in **a** we depict the dynamics as a function of the iterations, while in **b** as a function of the time, explaining the small displacement between both figures.

To better understand the localization properties of our process, we focus on the probability that an individual is active, sampled from the simulations. In Fig. 3a, we present the (hyper-)adjacency matrix, as in Eq. (1), while in Fig. 3b–d we show the individual probabilities extracted from branches I to III, respectively. The matrix is ordered according to the individual probabilities of Branch III (lower). This figure shows that Branch III (panel (d)) is constrained to a group of nodes (a community roughly defined as $C_1 = \{v_1, v_2, \cdots, v_{600}\}$), while Branch II (intermediate branch, panel (c)) is restricted to a different set of nodes together with some bridge hyperedges, and branch I accounts for the activation of all the nodes (see panel (b)). Here, we recall that bridges are defined as the hyperedges that are composed by nodes in different communities.

Figure 4 depicts the intermittent behavior observed in Branch III of Fig. 2 for the blues reviews hypergraph with $\lambda = 0.1016$. A similar behavior was observed for a range of parameters, but we choose this specific value of $\lambda$ for visualization purposes and to be consistent with the other figures. In Fig. 4a, we show the activity of the nodes as a function of the number of events (or iterations). The nodes are sorted by their activity for better visualization. Complementarily, in (b), we show the order parameter as a function of time. We observe that we have a set of nodes that are always active and a second set of nodes that can be activated due to some fluctuations. Comparing Fig. 4a with Fig. 3a, we notice that the

group of nodes in the upper-left corner of Fig. 3a are the most active ones. Note that they participate in a larger number of hyperedges (note the colors). On the other hand, the rest of this community (the remaining nodes of community $C_1$ in Fig. 3a) are the ones that have periods of activity and periods of inactivity.

Thus, the analysis at the individual level supports the initial hypothesis that intermittent behavior is a consequence of the activation and deactivation of a subset of nodes, or, in other words, the localization of states. The periods of high activity correspond to the activation of a sparser connected set of nodes by a more densely connected core. This latter core sustains the dynamics, and seeds the more sparsely connected nodes, which can only maintain its dynamics active for a limited time, and thus are responsible for the intermittent behavior.

## Multistability and intermittent behavior

The main results of the previous section were the existence of multistability and intermittency. Here, our primary goal is to provide further arguments to support these findings and to explain the mechanism behind these phenomena. Using the theoretical framework developed in section "Model definition and theoretical analysis" and the first-order approximation presented in the "Methods" section, we offer a strong argument in favor of our findings and against the possibility of them being simulation artifacts. Moreover, we propose a simple generative model for hypergraphs with community structure, which provides a possible mechanism for the observed phenomenology.

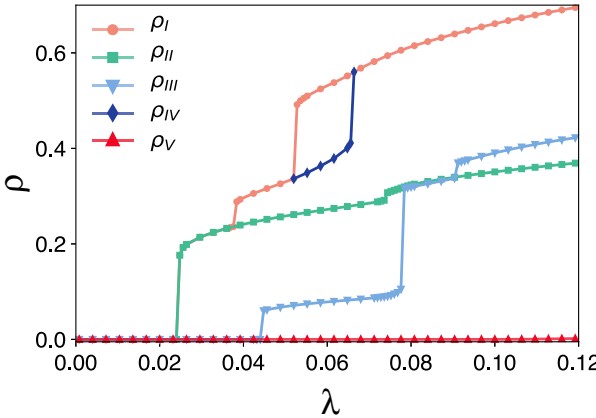

**Fig. 5 | Numerical solutions of the ODE system for the social contagion model in the blues reviews hypergraph.** We solved these equations using the same set of parameters as used in Fig. 2. For the Branch V, we considered an initial condition of $y_i(0) = 0.1$. The other branches were obtained considering the final micro-state of one point as the new initial condition for the next (both increasing and decreasing $\lambda$). The first point is obtained from our Monte Carlo simulations reported in Fig. 2.

Because of the finite size of the system, one may suspect the simulation to be trapped in metastates, that would vanish in a longer simulation. To counter this argument, we provide more robust evidence of multistability by providing numerical solutions of the ODE system in Eq. (5). In addition to strengthening our multistability argument, we also show that our first-order approximation is qualitatively correct in this specific scenario, which provides an additional argument that our approximation indeed captures the essence of our model.

We observed that our simulations have a strong dependency on initial conditions. So, to numerically solve the ODE system in Eq. (5) we used one micro-state obtained from our simulation as an initial condition. From this condition, we integrated Eq. (5) until reaching the steady-state. Finally, we used this solution as an initial condition to adjacent values of $\lambda$ (increasing and decreasing $\lambda$). With this algorithm, we were able to uncover the five branches shown in Fig. 5. We remark that, using an uniform initial condition, e.g., $y_i(0) = 0.1$ for all $i = 1, 2, \dots, N$, we were not able to find most of the branches in the ODE system. The exceptions are the absorbing state and the uppermost branch, which can be find using $y_i(0) = 1.0$ for all $i = 1, 2, \dots, N$.

Comparing Figs. 2b and 5, we can see a clear correspondence between the predicted (ODE) and observed peaks of the state distributions (MC). Because our approximation neglects correlations and fluctuations, we are not able to capture the behavior in Fig. 2a but only the peaks of the bimodal distributions. This comparison strengthens the argument that the observed multistability is not a simulation artifact but rather a genuine feature of the model. Note that our first-order approximation follows the same principles as the quenched mean-field in the SIS model in networks. In the network case, we only have unimodal distributions. Thus, this limitation is not an issue. However, in our case, further analysis is necessary, as we are not directly able to determine if the ODE's solution is a peak of a multimodal distribution or not. Finally, we also remark that typically the ODE overestimates the MC predictions slightly.

The analysis in section "Example of real-world hypergraphs: the case of blues reviews" suggests that the community structure in the blues hypergraph might be responsible for the multistability and the intermittent behavior. As noted in the previous section, Fig. 3b–d, different branches are related to different sets of nodes, thus suggesting localization. Complementary, for a visual argument, see, for instance, Fig. 3a, where we can see the block organization in the adjacency matrix. In this section, we explore this hypothesis by proposing an artificial model that captures the community structure without including other correlations (see the "Methods" section for a description of this algorithm). In this way, we can test the hypothesis that this type of structure is responsible for the observed dynamical behavior.

Figure 6 shows results for the QS Monte Carlo simulations in the artificial random model with communities, and changing the number of bridges, $m_{out}$, for values between $m_{out} = 200$ and $m_{out} = 600$. Although not shown, in all the cases, the absorbing state is stable and can be reached from a small initial condition (e.g., $\rho(t = 0) = 0.1$). From the first to the third column, we increase the number of bridge hyperedges, $m_{out}$, thus diluting the modular structure. For $m_{out} = 200$ (see Fig. 6a, d, g), we have multistability, as different initial conditions lead to different solutions. We also find a region in $\lambda$ where both coexist. In this case, the communities are sufficiently separated, and we do not observe intermittency. For $m_{out} = 400$ (see Fig. 6b, e, h), and $m_{out} = 600$ (see Fig. 6c, f, i), the multistability is not observed as different initial conditions led to the same solution. Interestingly, we observed intermittent behavior in the region between dashed lines in Fig. 6. In this region, we have a bimodal distribution of states and a susceptibility peak. Notice that, as we increase $m_{out}$ the susceptibility peak also moves, appearing for lower values of $\lambda$ (see Fig. 6h, i). More importantly, we recall that a similar behavior was observed in Branch III for the blues reviews hypergraph (see Fig. 2c), where we find a susceptibility peak caused by the intermittency.

For $m_{out} = 200$, we do not have a bimodal distribution. In this case, after the transition, we have two possible scenarios: one in which just one community is active, and a second one in which both communities are active. For $m_{out} = 400$ and $m_{out} = 600$, there is instead a region where a bimodal distribution is present. This distribution of states generates intermittent behavior due to the activation and deactivation of the sparser community, whereas the denser community sustains the process. However, the sparser one is only able to stay active for a limited time. During the lower activity periods, a strong enough fluctuation activates the sparser community. Nevertheless, after some time, this community will deactivate on its own due to another fluctuation.

These results suggest that when bridges are scarce, the communities are dynamically disconnected. Hence, we might have multiple stable solutions for a range of $\lambda$ due to localization. As we add bridging hyperedges, we allow the process to travel across communities. However, this can destroy the multiple stable solutions by merging them into a bimodal distribution of states and creating intermittency. We highlight that a similar effect was also observed by increasing/decreasing the hyperedge cardinalities and by changing the critical-mass threshold $\Theta^*$. In the first scenario, we noticed that by increasing the average hyperedge cardinality, we could change our system's behavior from multistability to intermittency. Particularly, by (i) considering the same artificial model with communities as in previous numerical simulations, (ii) fixing the number of hyperedges and bridges, but (iii) changing the average hyperedge cardinality, $\mu$, we were able to observe a shift from a multistable region for low $\mu$ to an intermittent behavior for larger $\mu$. Moreover, by changing the critical-mass threshold $\Theta^*$, we observed that, for higher values of $\Theta^*$, we tend to favor multistability, while for lower values of $\Theta^*$ we favor intermittency. The numerical simulations of changing $\mu$ and $\Theta^*$ are presented in the SI. It is worth highlighting that it might be possible to construct more complex hypergraphs that would display more branches and possibly even allow for multistability and intermittency at the same region of $\lambda$. Please see also the SI for an example with four communities. We remark that here we focused on the simplest structure that reproduces both phenomena. Furthermore, one can see a relation between our results and the previous findings[33] relative to the identification of network structures and of individuals best suited for spreading complex contagions. The authors proposed a centrality measure that

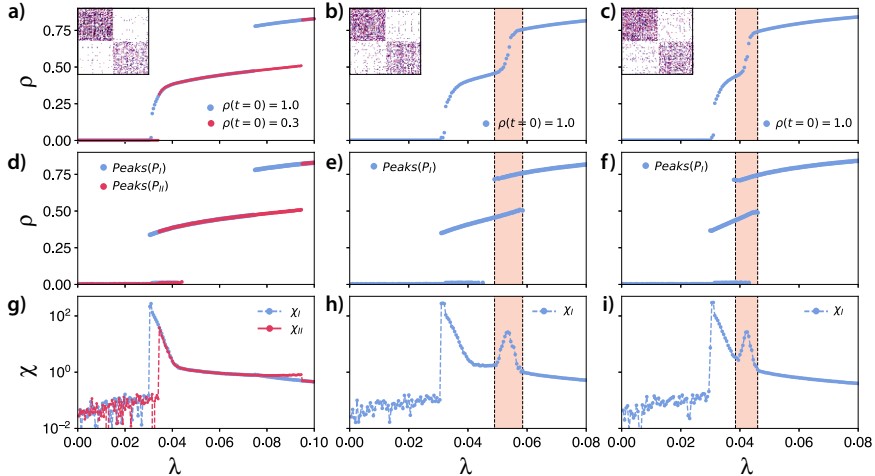

**Fig. 6 | Monte Carlo simulations for the social contagion model in artificial hypergraphs with community structure.** The simulations are organized by columns. From the first to the third column we show the numerical simulations for $m_{out} = 200$, $m_{out} = 400$ and $m_{out} = 600$, respectively. All the hypergraphs have $N = 10^3$ nodes and $n_c = 2$ communities. In **a**–**c**, we show the order parameter as a function of $\lambda$. In the top left corner of these plots, we show the adjacency matrix for each simulation. In **d**–**f**, we show the peaks of the state distributions. Finally, in **g**–**i**, we show the susceptibility plots. In **a**, **d**, **g**, we show this simulation for two different initial conditions, showing two possible solutions. In the other panels, the different initial conditions converged to the same solution. The dynamical parameters of our model are $\delta = 1.0$, $\Theta^* = 0.5$, and $\lambda^*(|e|) = \log_2(|e_j|)$.

accounts for the number of "enough wide bridges" between two nodes. Although in ref. [33] they are still using graphs (but the contagion is complex), this concept resembles the ideas behind critical-mass processes associated with our hyperedges. Thus, the term "enough wide bridges" might be understood as an abstraction of the critical-mass threshold in our context. We remark that the term "enough wide bridges" summarizes our results as it incorporates both the number of bridges (as shown in Fig. 6) and "how easy" it is to activate these bridges (results reported in the SI).

### Analysis of the transition between stable branches

As we increase or decrease $\lambda$, branches can become unstable, and the process might experience a transition from one branch to another. For disease spreading on networks, this transition is usually continuous. For example, consider an SIS process in an infinite, homogeneous network (thermodynamic limit). In this case, we have an absorbing state (disease-free state, $\rho^{SIS} = 0$) which is stable until the critical point. For any spreading rate larger than the critical point, the disease spreads through a collective activation of the network. In this regime, we have another branch that constitutes the active solutions ($\rho^{SIS} > 0$). This active branch "touches" the absorbing state at the critical point, making the transition continuous. However, when analyzing higher-order models, these transitions can be discontinuous[10,12,14]. Furthermore, here we observed that we might have multiple transitions for the same initial condition (see Fig. 2a). Despite this evidence, a complete characterization of these transitions is still lacking. In this section, we will focus our analysis on the nature of this transition, providing an argument supporting the hybrid nature of the transitions. In this class of transitions, we have discontinuity and scalings at the same time. We highlight that this proposition seems to be general as our finding explains all the observed behavior in the susceptibility curves not only in this paper but also the one reported in refs. [11,12].

To understand the nature of the transitions we study the hyperblob[12], which is a random regular graph, where every node has the same degree together with a hyperedge that includes every node. In this case, we can evaluate the exact distribution of states in the steady-state and observe how relevant quantities vary with system size. The graphical representation of the Markov chain that represents our dynamics in the hyperblob is shown in Fig. 7, where we already imposed the QS constraint, avoiding the absorbing state. In Fig. 8, we

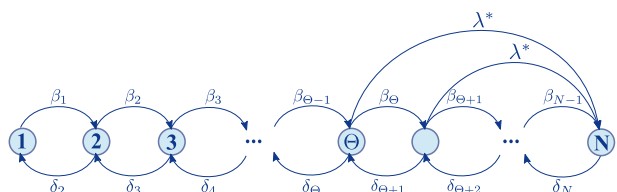

**Fig. 7 | Graph of transitions for the hyperblob respecting the QS constraint.** The graph illustrates the system's states through nodes, where each node corresponds to the number of active nodes. The arrows connecting the nodes represent the potential transitions between states. The rates for each transition are displayed above their respective arrow.

show the temporal behavior of our model, highlighting the importance of using the QS constrain as, for any finite system the dynamics will always converge to the absorbing state. Moreover, from this chain, we obtained the stationary distribution (see the "Methods" section for its derivation), which allow us to fully characterize our system in terms of the probability of having $n$ active nodes, $\pi_n$. From this quantity and, in addition to the order parameter and susceptibility, here, we are also interested in the probabilities that the number of active nodes is lower or higher than the threshold $\Theta$. The state with $\Theta$ active nodes is particularly important as, for $n \geq \Theta$, the Poisson Process $N_j^{\lambda}$ is created, which significantly changes our system's behavior. Formally, these probabilities are, respectively, expressed as

$$P_{Lower} = \sum_{j=0}^{\Theta-1} \pi_j, \tag{3}$$

$$P_{Upper} = \sum_{j=\Theta}^{N} \pi_j. \tag{4}$$

In Fig. 9, we show the order parameter, the susceptibility and the probability of each solution for $\lambda^* = 10$. For a complementary analysis of the hyperblob, please see Sec. III in the SI. As we found rapid changes in both the order parameter and susceptibility, its characterization in

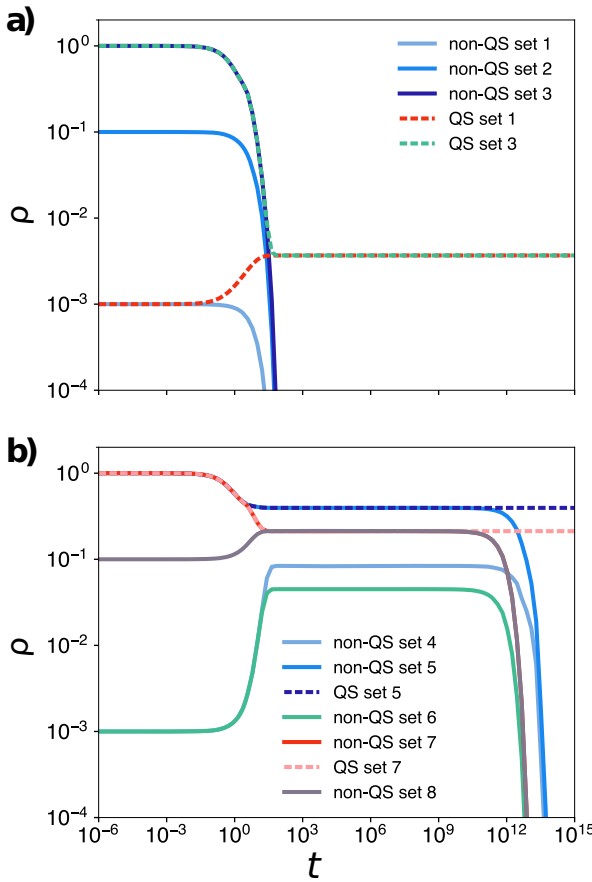

**Fig. 8 | Temporal evaluation of the exact model, Eq. (13) for different sets of parameters and initial conditions.** In **a**, the parameters are such that the dynamics converges to the absorbing state, while in **b** we have a metastate for the non-QS process and an active state for the QS constrained process. The hyperblob used in these simulations have $N = 10^3$ and the sets of parameters are: set $1 = \{\lambda\langle k \rangle = 0.9,$ $\lambda^* = 200, \Theta^* = 0.2, \delta = 1.0, \rho(0) = \frac{1}{N}\}$; set $2 = \{\lambda\langle k \rangle = 0.9, \lambda^* = 200, \Theta^* = 0.2, \delta = 1.0,$ $\rho(0) = \frac{100}{N}\}$; set $3 = \{\lambda\langle k \rangle = 0.9, \lambda^* = 200, \Theta^* = 0.2, \delta = 1.0, \rho(0) = 1\}$; set $4 = \{\lambda\langle k \rangle = 1.275,$ $\lambda^* = 200, \Theta^* = 0.2, \delta = 1.0, \rho(0) = \frac{1}{N}\}$; set $5 = \{\lambda\langle k \rangle = 1.275, \lambda^* = 200, \Theta^* = 0.2, \delta = 1.0,$ $\rho(0) = 1\}$; set $6 = \{\lambda\langle k \rangle = 1.275, \lambda^* = 200, \Theta^* = 0.3, \delta = 1.0, \rho(0) = \frac{1}{N}\}$; set $7 = \{\lambda\langle k \rangle = 1.275, \lambda^* = 200, \Theta^* = 0.3, \delta = 1.0, \rho(0) = 1\}$; and set $8 = \{\lambda\langle k \rangle = 1.275, \lambda^* = 200, \Theta^* = 0.3,$ $\delta = 1.0, \rho(0) = \frac{100}{N}\}$.

the thermodynamic limit can be achieved using these curves' left and right limits. These quantities are respectively denoted as $\rho^-$, $\chi^-$, and $\rho^+$, $\chi^+$. In practice, for the order parameter, $\rho^-$ ($\rho^+$) is defined as the first value that is larger (smaller) than $\frac{1}{N}$ from the lower (upper) solution. Complementary, for the susceptibility, we can use peaks in the derivatives of $\chi$ to define $\chi^-$ and $\chi^+$. We observed that, if $\lambda^*$ is low enough, the dynamics presents a second-order phase transition followed by a hybrid transition. Note that, by "low enough" we assume that $\lambda^*$ is constant for all sizes, i.e., do not scale with $N$, and it is not of the same order of the smallest size evaluated. These results are summarized in Figs. 9 and 10. In Fig. 9, we show the order parameter, the susceptibility and the probability of each solution for $\lambda^* = 10$. We notice a region where both solutions are possible, but only one solution exists for most of the evaluated parameters. The lower solution does not present any significant change compared to an SIS process in a homogeneous network. It exhibits a second-order phase transition, as shown in Fig. 9a and Fig. 10a, b, where we can see a diverging peak of susceptibility. As we increase $\lambda$, the system moves from the lower to the upper solution. A hybrid phase transition characterizes the transition between these two regimes. In this type of transition, we have discontinuities on the order parameter, a feature of a first-order phase

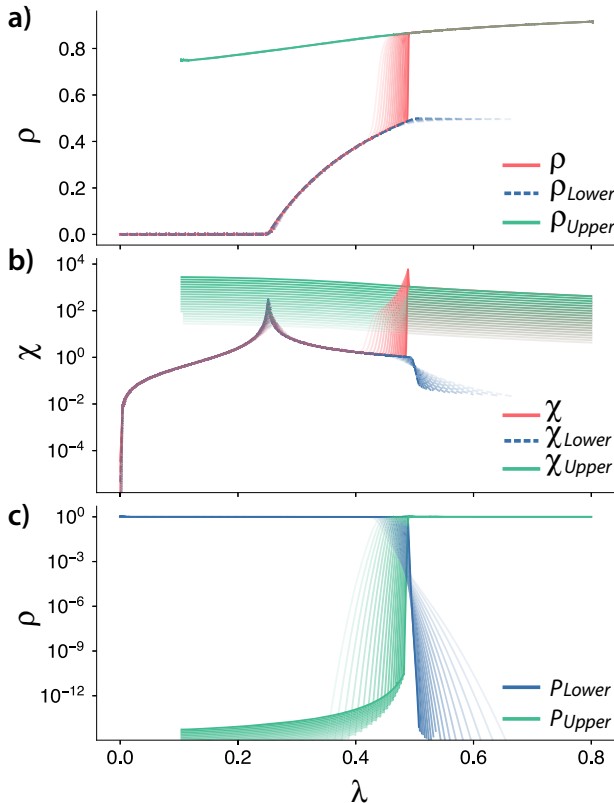

**Fig. 9 | Quantities of interest as a function of the spreading parameter $\lambda$.** For a "low" $\lambda^*$ (here, in this figure $\lambda^* = 10$), we have a second-order phase transition followed by a hybrid transition. We show the phase diagram in **a**, the susceptibility curves in **b**, and the probabilities of each branch in **c** for various system sizes (transparency corresponds to system size, i.e., the more transparent the curve, the smaller the system).

transition, and also scalings, which are a feature of a second-order phase transition[34,35]. We characterize this transition by showing that $|\lambda(\rho_2) - \lambda(\rho_2^+)|$ and $|\lambda(\chi_2^-) - \lambda(\chi_2^+)|$ tend to zero as we increase the system size, which is shown in Fig. 10e–h. The observed behavior implies that in the thermodynamic limit, we have a discontinuous transition. Importantly, the estimated exponent, $|\lambda(\rho_2) - \lambda(\rho_2^+)| \sim N^{-\mu}$, $\mu \approx 0.437 < 1$, satisfies the conditions for a hybrid phase transition. We also note that the susceptibility peak for the whole system, $\chi$, shows a diverging peak.

Interestingly, hybrid phase transitions were also found in a similar model for scale-free uniform hypergraphs[11]. Specifically, these results can be translated in our model by considering $\Theta_j = |e_j| - 1$. In other words, the model in[11] considers that the higher-order spreading processes will only be present if all the nodes but one are already active. Moreover, they are restricted to uniform hypergraphs. Nonetheless, these results are aligned with our findings, providing additional evidence that hybrid phase transitions might be common in higher-order systems.

## Discussion

A precise understanding of the dynamical properties of a model is fundamental for the correct observation, inference, and—possibly—control of the system. The expected behavior of social contagion models in simplicial complexes and hypergraphs are the discontinuous transitions and the emergency of a hysteresis cycle[10,12–14], which are not expected for processes in simple graphs[36,37]. Although these results were surprising on their own, here we showed that these models present an even richer phenomenology, including multistability, intermittency, and hybrid phase transitions. Our results also highlight the

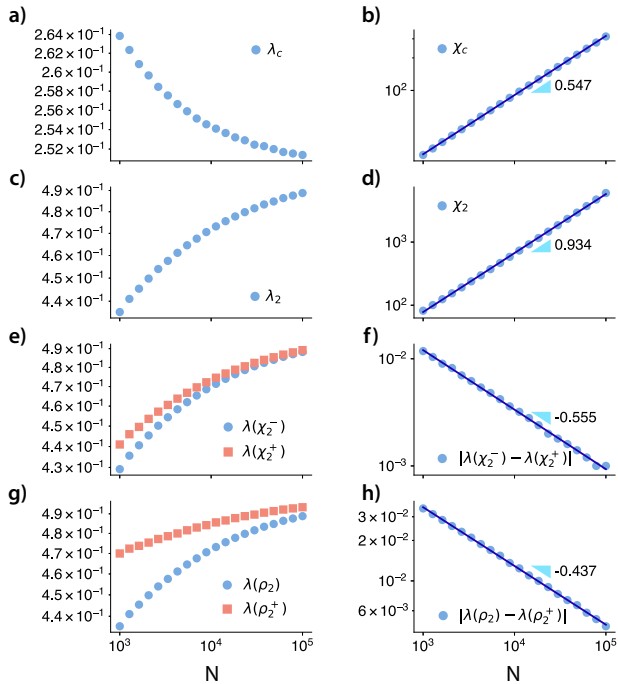

**Fig. 10 | Finite size analysis for the hyperblob.** For a "low" $\lambda^*$ (in this figure $\lambda^* = 10$), we have a second-order phase transition followed by a hybrid transition. We show the scaling of important quantities as a function of the system size. In this panel, from top to bottom, in **a**, **b**, we show the scaling of the lower solution, where $\lambda_c$ converges to a finite non-zero value while its respective susceptibility diverges, characterizing a second-order phase transition. In **c**, **d**, we observe that the susceptibility curve for the whole system also diverges. Besides, we can clearly see from **e** to **h** that $|\lambda(\chi_2^-) - \lambda(\chi_2^+)|$, in **e** and **f**, and $|\lambda(\rho_2) - \lambda(\rho_2^+)|$, in **g** and **h**, tend to zero as we increase the system size. The sub-figures from **c** to **h** are enough to characterize a hybrid transition. Note that in the thermodynamic limit, the transition is discontinuous as $\lim_{N\to\infty} |\lambda(\rho_2) - \lambda(\rho_2^+)| = 0$.

interplay between higher-order interactions and community structure. Although not universal, these are standard features in a wide variety of real systems and are particularly common in social contexts[38–40].

## Community organization might lead to localization of states
As observed in the real case and validated through artificial models, community structure in hypergraphs imposes dynamical localization of states. After a transition, the spreading can: (i) reach the whole population, while remaining delocalized; (ii) activate just a subset of individuals that scales with the system size, or (iii) activate just a node or a subset of nodes that does not scale with the system size. An example of the first scenario is the hyperblob, where the transition happens through a collective process, and all the nodes will be active with the same probability (for more details, see section "Analysis of the transition between stable branches" and ref. [12]). An example of the second one is instead observed in the case of the blues review hypergraph (see Fig. 3c, d, where the activity is constrained to a subset of nodes. Localization in community structured populations is not unexpected. However, the dynamic impact that it generates is indeed different from the graph cases. In graphs or multilayers we observe multiple susceptibility peaks associated to continuous changes in the order parameter[31,37,41], which contrasts with the phenomenology observed in our model and discussed next.

## Localization in higher-order models might generate intermittency, multistability, and/or multiple transitions
The localization in a subset of individuals, item (ii), might lead to multistability and multiple transitions between branches, as we

observed both in real data and artificial models (see Figs. 2 and 6). In this case, the branches are well separated, and the same set of parameters can activate different regions of the hypergraph, depending on the initial condition (see Fig. 6). This type of localization might also imply multiple transitions between stable branches. As an example, we can mention the solutions obtained by considering the initial condition $\rho(t=0) = 1$, either in the real hypergraph or in the artificial model with $m_{\text{out}} = 200$, Figs. 2 and 6 respectively. We observe two discontinuities, one separating the absorbing state and an active solution and another separating two activity levels. However, if we consider the artificial models with $m_{\text{out}} \geq 400$, we notice that the transition from the lower activity state to the higher activity state can also be continuous. Note that the concept of localization is not necessarily linked to multistability, as we might have localization with a single solution. Didactic examples are graphs with communities or multilayer networks. Here, our model reduces to an SIS in the graph scenario. In this case, we have a single absorbing state and localized processes, but the dynamics has a single accessible active state. This observation suggests that depending on the hyperedge size distribution (i.e., cardinality distribution), we might have localization without multistability. Note that, although the observed phenomena share some similar features with its graph[32,37] and multilayer[37,41,42] counterparts, here the mechanisms that guide localization and its macroscopic response are entirely different. In the pairwise case, the susceptibility and order parameter change continuously and, once a community is activated, it does not present abrupt temporal macroscopic variations. On the other hand, in the hypergraph case, we often observe significant macroscopic changes, which might be related to hyperedges intersections generating a cascading of activations. Moreover, in comparison with similar models on graphs (e.g., SIS), the social contagion model on hypergraphs displays a strong dependence on initial conditions. In fact, for a given set of parameters, the steady-state solution will depend mainly on the microscopic properties (e.g., localization of initial seeds) of the initial condition rather than on its macroscopic ones (e.g., total prevalence). For instance, in Fig. 2 we can see that, for the same macroscopic initial condition, depending on which community the initial seeds are placed, we reach a different branch. Moreover, we can observe the case in which a higher macroscopic initial condition leads to the absorbing state, while another with a lower macroscopic initial condition leads an active branch due to its micro-state configuration. Although not shown, we observed this behavior in most of our numerical simulations (see the "Methods" section for the algorithms employed to sample specific branches).

## Necessary and sufficient conditions for the observed behaviors
We were able to link the observed behaviors to the community structure. However, this does not imply that modular structures are the only ingredient able to generate multistability and intermittency. Indeed, other forms of structural correlation might play a similar dynamical role.

## The stability of the absorbing state
For an SIS in an infinity graph, the absorbing state will be unstable after the epidemic threshold, and we will have an active stable solution. In the hypergraph, the conditions are not as simple as in the graph case. If the intersections between hyperedges are smaller than the critical-mass threshold, activating one hyperedge is not enough to trigger a collective behavior, regardless of the spreading rate. Although we did not study the structural constraints related to this issue, they were verified during our simulations. This effect is also related to the role of the initial conditions in our process. For example, we can think of a uniform hypergraph as a line whose intersection between hyperedges is smaller than Θ. In this way, for a high spreading rate but a small initial seed, the process will fall into the absorbing state, implying that the absorbing state might be stable for a broader range of parameters. In

ref. [18], the stability conditions for the absorbing state and the critical point estimations were derived, already providing additional insights about this issue. However, further numerical simulations and the spectral analysis of hypergraphs might deepen our understanding about this process.

## Limitations

We must also point out the limitations of the methods employed here. We can not perform the finite-size analysis in most real systems, as we only have a single structure with a fixed size. In practice, this implies that we cannot precisely determine the phase transition type in these cases. However, through the analysis of both the order parameter and susceptibility, we obtain some understanding of these real systems. We showed that using Monte Carlo simulations and solving our ODE's (Eq. (5)) provides a more robust argument regarding the nature of a transition (continuous vs. discontinuous) and the existence and stability of multiple branches. We expect a peak in the susceptibility curve for hybrid transitions in real scenarios right after the discontinuity. This peak can be a sign of scaling behavior. As mentioned, this was observed for the hyperblob, the hyperstar, the exponential and power-law distributions of cardinalities in ref. [12]. From a theoretical viewpoint, measuring localization by only looking at the leading eigenvector of the adjacency matrix, as can be done in graphs, is not trivial, as we can not write the probabilities of activation as an eigenvector problem plus second-order terms. Although we have a visual indication that this matrix might encode some of the localization properties, further research is necessary to formalize this concept. Another limitation we identified is that our model does not incorporate backlash or cultural opposition, which is important from a sociological point of view. Indeed, we assume that the activation of a group increases the probability of activation of other groups. However, this might only be the case in some real scenarios. Such extension is left as a future work.

## Perspectives

We have shown that the social contagion model in hypergraphs presents a rich and unexpected behavior beyond its discontinuous transitions. In particular, we showed that, depending on the structure, we might have multistability and intermittency due to bimodal state distributions. Using artificial random models, we were able to show that this phenomenology can be associated with community structures in the hypergraph. Specifically, by controlling the number of bridges between two communities with different densities, we showed that fewer bridges create multistability, while the creation of bridges destroys multistability and induces intermittency. We highlight that although community structure is not a universal feature, it is still a widespread characteristic of real social systems. Moreover, it is possible that other structural ingredients could generate similar dynamical outcomes. As we have multiple branches, the importance of the transition between them also increases. Often we observe a discontinuity in the order parameter[10,12–14]. However, associated with this, we also have a divergence in the susceptibility, which is compatible with hybrid phase transitions. We formulated the exact equations for a hypergraph with structural symmetries, showing that the resulting dynamics indeed displays a hybrid transition. Although our argument is restricted to this specific structure, similar patterns were verified in all the simulations reported here, as well as in refs. [11,12], suggesting that hybrid transitions might be general.

We hope our results open new paths for the exploration of social contagion models in hypergraphs. Analytically, understanding the necessary and sufficient conditions for the observed phenomenology is one of the most challenging future problems. From a numerical perspective, the exploration and characterization of other real systems might also reveal so far unobserved behaviors as well as confirm our findings. Another view would be motivating further research about understanding the impact of our results on different processes. For instance, how can localization impact on synchronization of oscillators, diffusion, or opinion dynamics? Would we have multistability in such dynamics? Independently and concurrently to our study, multistability was also found in coupled oscillator systems with higher-order interactions and community structure[43]. This findings reinforce our conjecture that such phenomenology might be common in higher-order interactions.

Our findings might also impact the design of real experiments similar to the ones in refs. [5,44,45]. One of the main difficulties with this type of experiment is that the number of people participating is often reduced, and the signals in the observables are usually noisy. In such small systems, while accurately measuring multistability might be challenging, intermittency might be easier to capture as we would be interested in finding periods of high activity followed by periods of low activity. Along similar lines, data coming from online social systems, while abundant in volume and number of potential subjects, is less controlled, imposing limitations on the modeling possibilities. Despite these limitations, there are still many available datasets that are higher-order in nature (i.e., the most natural representation would be a group and not a collection of pairwise interactions), for instance, WhatsApp message exchange in groups (see refs. [46,47]) or data from Reddit as the collaboration in the social experiment r/place[48]. We remark that, in principle, studying these datasets from the viewpoint of higher-order interactions is possible. However, this task is not trivial, and we left them as future work. Another foreseeable future direction would be incorporating different mechanisms as variants of the original model. For instance, one might propose variations that solve some of the above-mentioned limitations, e.g., including backlash or cultural opposition. Another possibility would be a variant that explicitly considers Alport's contact hypothesis[49].

To conclude, the literature on threshold models suggests that many processes can be modeled as binary choice critical-mass processes. For example, in ref. [2], the author proposes a catalog of processes that includes diffusion of innovation, rumors and diseases, strikes, voting, educational attainment, leaving social occasions, migration, and experimental psychology. We must highlight that in ref. [2], the author associates the threshold processes to the individuals and not the groups. However, the threshold is reached or not due to individual social interactions. Our approach is slightly different as we focus on the group rather than the individuals. Despite these differences, the proposed catalog is still valid in our case. The main difference is that our model might provide different mechanistic explanations for similar phenomena. We should also complement the argument for the case of disease spreading following a similar reasoning as in ref. [17]. We presume that our model may provide new insights into a disease spreading in which there is a viral load threshold[50]. Since, in this case, sharing an environment with a few infected people might impose an increased risk higher than linear, which would be the standard complex network prediction, our model could better explain this process. Finally, we could also mention examples from our daily lives that can be conjectured as a result of group interactions. For example, some of the phenomena described by Malcolm Gladwell in his book, *Tipping point: How Little Things Can Make a Big Difference*[51], can also be interpreted or re-analyzed from the group dynamics point of view. A notable example would be the famous saying that "fashion is cyclic" is an effect of group interactions as fashion can be understood as a norm, as in refs. [5,24]. In this scenario, we hypothesize that the observed cyclic behavior is associated with the structural organization of our societies.

## Methods

### The first-order approximation (individual-based)

Equation (2) expresses the exact process, however, it cannot be numerically solved due to its computational cost. Notice that we need $O(2^N)$ equations to exactly solve this system. Thus, assuming that the

random variables are independent and denoting $y_i = \langle Y_i \rangle$, we obtain the first-order approximation as

$$\frac{dy_i}{dt} = -\delta y_i + \lambda(1 - y_i) \sum_{e_j \in \mathcal{E}_i} \sum_{k=\Theta_j}^{|e_j|-1} \lambda^*(|e_j|) \mathbb{P}_{e_j}^{v_i}(K = k), \qquad (5)$$

where we assumed that the spreading rate is composed by the product of a free parameter and a function of the cardinality, i.e., $\lambda_j = \lambda \times \lambda^*(|e_j|)$ and $\mathbb{P}_{e_j}(K = k)$ is the probability that the hyperedge $e_j$ has $k$ active nodes inside. Specifically, we estimated the expectation of the indicator function as a Poisson binomial distribution. Formally,

$$\left\langle \mathbb{1}_{\{Y_i = 0, T_j \geq \Theta_j\}} \right\rangle \approx (1 - y_i) \sum_{k=\Theta_j}^{|e_j|-1} \mathbb{P}_{e_j}^{v_i}(K = k), \qquad (6)$$

$$\mathbb{P}_{e_j}^{v_i}(K = k) = \sum_{A \in F_k} \prod_{m : v_m \in A} y_m \prod_{\ell : v_\ell \in A^c} (1 - y_\ell), \qquad (7)$$

where $F_k$ is the set of all subsets of $e_j \backslash \{v_i\}$ with cardinality $k$, $A$ is one of those sets, and $A^c$ is its complementary, i.e., the remaining nodes of $e_j$ who are not $v_i$. Intuitively, $A$ accounts for the active nodes of each possible single microstate and $A^c$ for the inactive ones. Notice that combining $A$ and $A^c$ represents $|e_j| - 1$ nodes as $v_i$ is excluded. Thus, the summation over $F_k$ considers all possible micro configurations in a given hyperedge. In Fig. 1 (a) we show an example of the possible micro states for each hyperedge and their associated value of $T_j$. Unfortunately, Eq. (7) is not numerically stable if $|e_j|$ is large. Note that, calculating $\mathbb{P}_{e_j}(K = \ell)$ using Eq. (7) involves the multiplication of $|e_j|$ terms that are smaller than one. Thus, for a large $|e_j|$ we might have underflow issues. It is however possible to stabilize its solutions by considering the discrete Fourier transform[52]

$$\mathbb{P}_{e_j}^{v_i}(K = k) = \frac{1}{|e_j|} \sum_{l=0}^{|e_j|-1} C^{-lk} \prod_{\substack{m \, : \, v_m \in e_j; \\ m \neq i}} \left(1 + (C^l - 1)y_m\right), \qquad (8)$$

where $C = \exp\left(\frac{2\pi \mathbf{i}}{|e_j|}\right)$, where $\mathbf{i}$ is the imaginary unit. Note that node $v_i$ is excluded here, and the normalization should also change accordingly. We also remark that this approach allows us to compute the solution for arbitrarily large hyperedges. Thus, we can numerically solve the first-order approximation in Eq. (5) using the approximation in Eq. (8). Note also that, Eq. (6) is an approximation as we assume that the nodes' state is independent. However, Eqs. (7) and (8) are exact for independent random variables and are also identical, giving the same results.

The ODE solutions were implemented using the Gnu Scientific library[53]. More specifically, we used the explicit embedded Runge–Kutta–Fehlberg (4, 5) method, with an adaptive step-size control, where we keep the local error on each step within an absolute error of $\epsilon_{abs} = 10^{-4}$ and relative error of $\epsilon_{rel} = 10^{-3}$ with respect to the solution $y_i(t)$.

### Continuous-time simulations

We want to both validate the expressions developed in the previous sections, and statistically describe our model in arbitrary hypergraphs. To achieve this, we use continuous-time Monte Carlo simulations, more specifically, we use the Gillespie algorithm[54], which can be described as follows. First, we create a vector containing the times associated with all possible Poisson processes. As they are Poisson processes, the inter-event times are sampled from an exponential distribution with the appropriate parameters. For instance, if it is a deactivation process, the exponential distribution has parameter $\delta$. If

the process is associated with a spreading, the parameter will be $\lambda \times \lambda^*(|e_j|)$. If the process is not active, we set it as $\infty$ (effectively the largest double). Thus, given an initial condition, the dynamics run on top of this vector of times. On each iteration, we find the element with the shortest time and execute its associated rules, which can be deactivation or spreading. Note that new processes might be created or deleted accordingly. For example, if a hyperedge reached its critical mass, the Poisson process for that event will be created. However, if, before its execution, a sufficient number of nodes is deactivated (making the hyperedge stays below its critical mass), the process should be removed. Next, our time variable is increased according to the time associated with the executed Poisson process. The same rules are repeated until reaching the absorbing state or a $t_{max}$. This algorithm was initially proposed in ref. [12], and it is an extension of the methods described in Section 10.3 of ref. [37].

### Quasi-stationary method (QS)

Our model has a single absorbing state, the state in which every node is inactive. So, for any finite system with finite rates, the dynamics will reach this state. Mathematically this can be avoided by restricting our process to active states (see section "Quasi-stationary steady-state solutions"). Computationally, we adopt a similar approach. We avoid the absorbing state by moving to a previously visited activate state every time the system falls in the absorbing state. The algorithm is defined as follows. We keep a list of $M$ previously visited active states. This list is continuously updated. If we are in an active state, with a probability $p_r \Delta t$, the current state replaces a random position of this list. If the absorbing state is reached, then a random state in the list replaces the absorbing state. We let the dynamics relax for $t_r$ and, after that, during a time $t_s$, we sample the distribution of states, $\mathbb{P}(n)$, where $n$ is the number of active nodes. Note that, on each iteration of the described algorithm, we are computing Freq($n$) ← Freq($n$) + $\Delta t$. In other words, we are computing the time our dynamics spent in the state $n$. Hence, $\mathbb{P}(n) \propto$ Freq($n$). From that, we characterize our dynamics using the order parameter and the susceptibility, respectively defined as

$$\rho = \frac{\langle n \rangle}{N}, \qquad (9)$$

$$\chi = \frac{\langle n^2 \rangle - \langle n \rangle^2}{\langle n \rangle}. \qquad (10)$$

This method was initially proposed in ref. [55] and had been extensively used in the analysis of epidemic spreading[31,37,56,57].

We remark that $t_r$ and $t_s$ vary according to the system size, and the algorithm is stable to the choices of list size $M$ and probability $p_r$. To reduce the computational cost of this method, we also employed an adaptive version. In this version, we define a variable sampling time given as $t_r + ct_s^*$, where $t_s^*$ is a smaller time-window and $c$ is not set but defined by the convergence of $\chi$. In practice, we calculate $\chi$ before and after each $t_s^*$ time-window. If the absolute difference between the susceptibility is lower than $\epsilon$ (here set as $\epsilon = 0.001$), the algorithm stops. Additionally, we also define a $c_{max}$ (here set as $c_{max} = 500$), which is the stop condition. Thus, we expect to reduce the computational cost with this adaptive version while keeping statistically reliable measurements.

Moreover, as we have bimodal distributions, aside from the order-parameter, $\rho$, and the susceptibility $\chi$, it is also necessary to keep track and store the state distributions, $P(n)$. We will be interested in looking at the multiple peaks of these distributions, especially the value of $\rho$ at which these peaks appear. Here, this quantity is denoted as $Peaks(P)$. Notice that, in the single-mode case, the peak represents the most likely value.

## Multistability and simulation methods

As shown in the main text, our model strongly depends on the initial micro-state, which might generate bimodal distributions or multiple stable branches for the same parameters. In the bimodal distribution case, we have an intermittent temporal behavior, and the main challenge, in this case, is sampling for long enough. Additionally, when the probability of moving from one branch to the other is very low (and not found in our numerical simulations – it will be zero only in the thermodynamic limit), we have multiple stable branches for the same value of parameters. In this case, the difficulty is finding the initial condition that will lead to such a solution. Thus, to properly explore our parameter space, we employ a two-step process. First, we explore different random initial conditions for a series of parameters, revealing some branches. They will be visible as a concentration of points in some regions of the $\rho \times \lambda$ diagrams. Next, to properly sample the already found branch, we use similar initial conditions to obtain the complete branch. We cannot guarantee that a given simulation will reach the expected branch due to stochastic fluctuations and the initial condition dependency. Thus, to circumvent this problem, we need to run many simulations using different initial conditions and discard those that fall in branches we are not interested in. With this process, we can sample from different branches. We remark that this procedure might be costly as we have no guarantee that the chosen initial condition will arrive at the desired branch. Despite that, in practice, this method gives reasonable results as it allows us to explore the parameter space without introducing any bias in the found solutions.

Alternatively to the random initial conditions, we can also use our knowledge of our structure and set specific initial conditions. As observed in the main text, communities are reasonable candidates to sustain the activity and, macroscopically, generate a stable solution. So, the alternative algorithm is to use as an initial condition one or more communities as active and the remaining communities as inactive. This approach was used in the Supplementary Information, sections IV and V, while the exploration of different random initial conditions was used in the main text. Naturally, this method can be extended to any initial condition of interest. We highlight that the alternative approach reduces the computational cost when we have some knowledge about our system. However, we might also be less likely to sample all the branches due to the introduced bias on the initial condition. Note also that it is very difficult to guarantee that we found all the possible branches for a given system and set of parameters.

## Artificial hypergraph model

Here, we propose a hypergraph extension of the community structure model presented in ref. [38]. The algorithm is described as follows. The number of nodes, $N$, and communities, $n_c$, is fixed. The hyperedge cardinalities will be sampled from a fixed distribution, $P(|e_j|)$. For each community $c$, we have $m_{in}^c$ hyperedges that will be constructed using only nodes inside the community. Each community can have a different density. To link two different communities, we have $m_{out}$ hyperedges that will constitute the bridges. In this case, we extract a uniform number from $\ell \in [1, |e_j|)$, where $\ell$ is the number of nodes in one community and $\ell - |e_j|$ will be in the other community.

In our numerical simulations in the main text we used $P(|e_j|) = \text{Exp}(\mu)$ with $\mu = 8$ but imposing that $\min(|e_j|) = 2$ and $\max(|e_j|) = \frac{N}{n_c}$. For simplicity, we build a hypergraph with $N = 10^3$ nodes organized in $n_c = 2$ communities. The community parameters are $m_{in}^1 = 1000$ and $m_{in}^2 = 500$, creating different levels of activation for the different groups. Finally, we leave $m_{out}$ as a free parameter to control the number of bridges, aiming to observe and control the dynamical behavior of our model.

## Exact equations for the hyperblob

In general, our exact formulation in Eq. (2) cannot be analytically solved for an arbitrary hypergraph. Nonetheless, by considering a homogeneous hypergraph, we can reduce the complexity of the problem and still calculate exact quantities. Henceforth, we focus on the so-called hyperblob[12]. This hypergraph is defined as a set of homogeneous pairwise relationships, forming a random regular graph, where every node has $\langle k \rangle$ edges, together with a hyperedge containing all the nodes. As the nodes are indistinguishable by their degree we can describe the state of our system by the number of active nodes $n$. Thus, the transition rates can be expressed as

$$
\begin{cases}
\mathbf{Q}_{n,n-1} = \delta n = \delta_n \\
\mathbf{Q}_{n,n+1} = \lambda \langle k \rangle n \frac{(N-n)}{N} = \beta_n \\
\mathbf{Q}_{n,N} = \lambda^* U(n - \Theta)
\end{cases}
\quad , \tag{11}
$$

where $U(n - \Theta)$ is the Heaviside step function and the element $\mathbf{Q}_{i,j}$ is the transition rate from the state with $i$ active nodes to a state with $j$ active nodes. The elements that are not explicitly defined in Eq. (11) are zero. Note that, if $\lambda^* = 0$ we recover an SIS dynamics in an homogeneous population. Figure 7 is a graphical representation of these transitions but restricted to active states (see section "Analysis of the transition between stable branches" in the main text). Consequently, we can express the temporal evolution of our dynamics as

$$
\frac{dP}{dt} = \mathbf{Q}^T P, \tag{12}
$$

where $P = [P_0, P_1, \ldots, P_N]^T$ is a vector whose elements $P_n$ are the probabilities of having $n$ active nodes. This equation can be solved as

$$
P(t) = \exp(\mathbf{Q}^T t)P(0). \tag{13}
$$

Moreover, denoting the steady-state solution as $\pi \in \mathbb{R}^N$, it can be obtained as

$$
\pi = \text{Null}(\mathbf{Q}). \tag{14}
$$

## Quasi-stationary steady-state solutions

For any finite hypergraph, the only absorbing state in our dynamics is the state $n = 0$. Consequently, regardless of the parameters of our dynamics, we will always reach this state. However, for sufficiently large hypergraphs, the dynamics will arrive at a metastate and remain there for some time. After leaving this state, the system will reach the absorbing state. In general, we are interested in the metastate instead of the absorbing state. So, to obtain insights about this state, we use the quasi-stationary distribution, which is constrained to active states. Computationally, this is effectively implemented by the QS method, described in section "Multistability and simulation methods." Mathematically, this is done by imposing that the transition rate to this state is zero. As the process is defined in continuous time and the probability of two events happening at the same time is zero, we can implement the QS constraints as

$$
\mathbf{Q}_{1,0} = 0. \tag{15}
$$

A graphical representation of the QS-constrained chain is shown in Fig. 7. Moreover, Eqs. (12) and (13) are also valid after applying the QS constraint, Eq. (15). Note that, without this constraint, the process depends on the initial condition, while the QS-constrained system does not.

Under the QS constraint, Eq. (12) is expressed as

$$\frac{dP_0}{dt} = 0 \tag{16}$$

$$\frac{dP_n}{dt} = \delta(n+1)P_{n+1} + \lambda\langle k\rangle(n-1)(N-n+1)P_{n-1} + \\ - \left[\delta n + \lambda\langle k\rangle n(N-n) + \lambda^* U(n-\Theta)\right]P_n \tag{17}$$

$$\frac{dP_N}{dt} = \lambda\langle k\rangle(N-1)P_{N-1} + \lambda^* \sum_{i=\Theta}^{N} P_i - \delta N P_N, \tag{18}$$

where $P_n$ is defined for the interval $n \in [0, N]$ and the limits are explicitly shown. In the steady-state, i.e., $\frac{dP_n}{dt} = 0$ for all $n \in [0, N]$, we can analytically obtain the stationary distribution as

$$\pi_0 = 0 \tag{19}$$

$$\pi_1 = \frac{\delta 2\pi_2}{\lambda\langle k\rangle(N-1) + \lambda * U(1-\Theta)} \tag{20}$$

$$\pi_{n+1} = \frac{\left[\delta n + \lambda\langle k\rangle n(N-n) + \lambda * U(n-\Theta)\right]\pi_n}{\delta(n+1)} + \\ - \frac{\lambda\langle k\rangle(n-1)(N-n+1)\pi_{n-1}}{\delta(n+1)} \tag{21}$$

$$\pi_N = \frac{\lambda\langle k\rangle(N-1)\pi_{N-1} + \lambda * \sum_{i=\Theta}^{N} \pi_i}{\delta N}, \tag{22}$$

where the normalization $\sum_{i=0}^{N} \pi_i = 1$ must be respected. Although we can not obtain a closed expression for $\pi$ for a fixed size and set of parameters, we can calculate its exact distribution of states. The computational cost of this calculation is $O(N)$, which allows us to evaluate reasonably large systems.

In Fig. 8, we show two examples of temporal behaviors by solving Eq. (13) with the appropriate matrices. In Fig. 8a, the system is below the critical point, while in Fig. 8b the dynamics operates above it. In the non-QS case, below the critical point, $\rho$ goes exponentially fast to the absorbing state ($\rho = 0$), while in the QS-constrained case, it goes to a state near $\rho \approx \frac{1}{N}$. Above the critical point, Fig. 8b, we can observe that $\rho$ stays at the metastate before converging to the absorbing state. Moreover, we can see the dependency on the initial condition, where, for the same set of parameters but different initial conditions, the dynamics has a different metastate. For an example, compare Fig. 8b, curves for set 4 and set 5. Also, note that the respective QS-constrained system converges to a state compatible with the non-QS set 5. Intuitively, the differences in solutions for sets 4 and 5 are related to the probability of getting to the absorbing state due to finite-size fluctuations. For set 4, this is evident as the process begins with a single active node.

### Reporting summary
Further information on research design is available in the Nature Portfolio Reporting Summary linked to this article.

## Data availability
Most of the data used in our manuscript are artificially generated by computational simulations whose methods are explained in the text. The blues reviews hypergraph is publicly available at https://www.cs.cornell.edu/~arb/data/. The data are available from the corresponding author upon request.

## Code availability
The algorithms used in our numerical simulations are described in the "Methods" section. Custom code is implemented in C/C++ and can be found at https://gitlab.com/guifarruda/socialcontagion.

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

## Acknowledgements

G.F.d.A. and Y.M. acknowledge the financial support of Soremartec S.A. and Soremartec Italia, Ferrero Group. G.P. acknowledges partial support from Intesa Sanpaolo Innovation Center. Y.M. was partially supported by the Government of Aragón, Spain and "ERDF A way of making Europe" through grant E36 20R (FENOL), and by Ministerio de Ciencia e Innovación, Agencia Española de Investigación (MCIN/AEI/ 10.13039/501100011033) Grant No. PID2020 115800GB I00. The authors acknowledge the use of the computational resources of COSNET Lab at Institute BIFI, funded by Banco Santander through grant Santander UZ 2020/0274 and by the Government of Aragón (FONDO-C19-UZ-164255). The funders had no role in study design, data collection, and analysis, decision to publish, or preparation of the manuscript.

## Author contributions

G.F.d.A., G.P., and Y.M. conceived and designed the study; G.F.d.A. performed the experiments; G.F.d.A., G.P., P.M.R., and Y.M. analyzed, discussed the results, wrote the paper, and contributed to the revision of the final manuscript.

## Competing interests

The authors declare no competing interests.
