## [Peer Review File · Nature Communications]

REVIEWER COMMENTS

Reviewer #1 (Remarks to the Author):

This manuscript explores the higher-order dynamics introduced in Ref. [10] and discover multistability, intermittency, and hybrid transitions in the real-world synthetic hypergraph. The authors investigate the origin of such phenomena and show that the phenomena can be at least partially explained by the community structure of the hypergraph. The manuscript shows that the strength of the connection between the communities, which is characterized by the number of bridges (even though bridge was not clearly defined in the manuscript, which is necessary before publication), is closely related to the phenomena and concludes with the implications of the results to various social systems: Fewer bridges create multistability, while the creation of bridges induces intermittency.

The paper tackles one of the most vibrant research topics in the field to present interesting and novel results. Methods are clearly illustrated and appropriate for the research. Thus, I deem the manuscript suitable for publication in Nature Communications. However, to gain my definitive approval, the author should address the issues listed below:

- “Bridge” seems to be an important quantity in the manuscript. Nevertheless, the quantity was not clearly defined. It may be good to define it in Fig.1 and related content.
- The methods used to determine which branch a particular simulation belongs to should be explained. If I understand correctly, two modes are considered to be in separate branches if there is zero probability of transition between them. It should be emphasized that transitions between the branches do not occur, or at least never occurred in the numerical simulations in this research.
- Caption of Fig. 3: The caption states the adjacency matrix is ordered according to the activity of Branch I. Panels (b), (c), and (d) seem to be ordered according to the activity of Branch III. The main text states that the matrix is ordered according to the individual probabilities of Branch III. The orderings of the nodes in the figure need to be clarified.
- The word “experiment” in the manuscript can be misleading. It should be replaced with “numerical simulation” or another equivalent word that accurately describes the work of this research.
- Page 3: This happens at the first time of the Poisson processes, which means that as soon as the threshold is hit, the inactive vertices become activated instantly after a random time exponentially distributed with parameter λ_i .

⇒ It is not clear what the phrase “first time of the Poisson processes” means.

⇒ “as soon as” and “instantly” are contradictory to “after a random time exponentially distributed”. If I understand correctly, the word “instantly” should be replaced with “simultaneously”. Moreover, the contagion event does not happen as soon as the threshold is hit. Maybe only the time of the contagion

event is determined in the numerical simulation algorithms as soon as the threshold is hit. It should be clarified.

- Page 6:

“the “frequency” at which the system switches between the two modes changes the variance and, therefore, the susceptibility.”

⇒ the relative time the system spends in each mode, rather than how often the transitions between the modes occur, determines the variance.

- On page 14: A very strong conclusion has been drawn with very little basis for the “pro-woman” laws--example. Either a stronger basis has to be presented, the claim has to be softened, or the example has to be removed from the manuscript.

A list of minor comments also follows:

- In the caption of figure 2, “In (a), (b) and (c), we present the order parameter, susceptibility, and the peaks” should be reordered as “In (a), (b) and (c), we present the order parameter, the peaks, and the susceptibility”

- On page 2, the indicator function needs to be defined, because it is not a commonly used symbol in the research community.

- On page 3, the symbol for the ceiling function should be introduced in the text, because it is not a commonly used symbol in the research community.

- In eq. (2), the condition on δY_i is written first in the indicator function, while in the text, the condition on δT_j is written first. They should be equalized.

- It is more appropriate that λ^* is introduced in Sec. II A rather than Sec. II B.

- On page 4, “a random element of the list” is better replaced with “a random state in the list”

- In figure 6 (a) and (d), it is impossible to distinguish where the blue dots are overlapped by red dots unless the figure is highly magnified. Because distinguishing it is important to interpret the figure, this problem should be resolved (Maybe by introducing transparency to the dots).

- On page 10, “these transitions are discontinuous” → these transitions can be discontinuous

- In the caption of figure 9, “here, in this figure” → “here,” or “in this figure,”

- On page 10, it should be stated that the edges form a random regular network in the hyperblob.

- On page 10, “As the nodes are indistinguishable” → Nodes are distinguishable in random regular networks.

Reviewer #2 (Remarks to the Author):

This paper studies the dynamics of critical mass in collections of interconnected groups.

1. The authors adopt a computational (simulation-based) approach to the study of critical mass, and suggest that their approach to studying inter-group dynamics is novel. However, unbeknown to the authors, there is an enormous literature in social psychology, political science, and other fields that examine intergroup dynamics in large scale behavior change, starting with Allport's (1954) contact hypothesis, and extending through several decades of literature challenging it and extending it.

2. The authors should also be aware that inter-group dynamics within a society or population often involve political and cultural divides across groups. Thus, critical mass within one group does not necessarily translate into critical mass within other groups, and may instead result in backlash, or cultural opposition from other groups.

3. I believe the authors are aware of the differences between empirical studies and simulation-based studies, however they note in the introduction to their article, that their model, "provides a theoretical foundation for, and phenomenological explanation to, the seemingly different experimental findings of expected critical mass thresholds. Studies based on a single group suggest a threshold between 30%-40%. Conversely, a critical mass of 10% would correspond to a population composed of groups of diverse sizes, each one with a different threshold."

This is a problematic motivation for their study since it incorrectly characterizes the literature. The 10% result (Xie et al. 2011) refers to a simulation model based on the naming game, in which one population is overtaken by multiple 'memes' competing for dominance. Experimental results (Centola et al. 2018) show a critical mass of 25% for empirical populations engaged in critical mass dynamics, and provide a mathematical model of these dynamics. And, ethnographic studies (Kanter 1977) show a critical mass of around 25%-30%, but without analytical data. The second and third studies are both empirical and largely consistent with one another, while the first is a simulation with no empirical foundation. I do not see that there is a tension in the literature here, particularly since the present study is a simulation model without empirical grounding, so it is unclear how it will add to the current empirically grounded literature.

4. The authors present their work as "a dynamical analysis of the social contagion model presented in Ref.[10]." There is an enormous variety of social contagion models, many of which have solid empirical foundations. I cannot understand why this article selects the model in Ref.[10] as the basis for its analysis. It does not seem to have much grounding or connection to the broader literature on contagion, but is a quite narrow model, which this paper then elaborates in more detail. This kind of modeling exercise does not seem suitable for a general interest scientific journal. Consistent with this assessment, I find the analysis of the 'Blues Reviews' hypergraph completely un-compelling since this seems like a fitting exercise, and it provides no substantive or micro-foundational model that would motivate the

applicability of their particular theoretical approach to these data (nor would it explain why their theoretical approach would be expected, or not expected, to work for other kinds of data).

5. The relationship between bridges, or in other words, “connectivity”, and multistable states is well-known, as it is essential for coordination across communities that those communities are sufficiently connected.

6. The analyses provided in this study vary between i) confirming existing theoretical intuitions about dynamics of coordination in graphs, and ii) being so specific to the hypergraph-theoretic architecture upon which this study is based that the counterintuitive results are not convincing in light of the article's claims of generality.

Reviewer #3 (Remarks to the Author):

See attached report file

Review report NCOMMS-22-14404

The authors study the behavior of a social contagion model that they previously proposed. They find a rich set of behaviors that is different than that usually observed in standard pairwise graphs and they associate it with community structure. The results are interesting, in particular the rich phenomenology of intermittency and metastability. The paper is clear and well written. I think the paper makes a relevant contribution and deserves publication. I have few comments for suggestions to make it even stronger, as I describe below.

Major points

1. The paper is quite long and dense of results. While there are several results that the authors would like to present, I think it is worth prioritizing and shift some of them into the SI for better highlighting the most important ones. For instance, I believe that the description of the hyperblob case in Sec V could be condensed. Many details of the calculations can go in the SI and they could compress that section by focusing on the main results instead.
2. The real data they used is very sparse, $N = 1106$ and only 694 hyperedges. I am wondering if results would change for sparsity that is still sparse but higher than this quite low value. Maybe results would be less sensitive to initial condition? Maybe we see less branches? I would also recommend to report few more statistics about this datasets, e.g. the distribution of $|e_j|$, node degree distribution and something describing how many lower size hyperedges are already contained into bigger ones (e.g. how many pairs or triangles are contained into hyperedges of $|e_j| > 2, 3$, to measure their redundancy).
3. Similarly, I would be interested to understand how results are impacted by the distribution of hyperedge sizes. For instance, one could check if we see similar behaviors when varying the max hyperedge size. I expect that when this is small, e.g. 3-5, we get a more similar picture to the pairwise case. So it would be interesting to see if there is a critical value after which, e.g., intermittency or multistability is observed. My intuition is that what makes the behavior rich and different from the pairwise graph case is the overlap between hyperedges. This becomes more important as hyperedges become bigger. It may also depend on θ^* . Hence this type of analysis should be investigated.

Minor points

1. How many rewiring did you consider in the randomly rewired version of Fig.2 ? I cannot see error bars to understand whether there were few instances where the rewiring could lead to behaviors similar to the real case.
2. It is not clear if the results on the Blues reviews are obtained using only the giant connected component accounting for the hyperedges. They mention the pairwise graph giant component of 24 nodes, but is not clear if accounting for hyperedges we obtain a unique component of 1106 nodes. Please clarify this.
3. Is Eq 6 an approximation as written in the text? In this case I would add change the symbol $=$ with the symbol \approx . As this equation comes from a reference it is hard to tell if it exact or it involves some kind of approximation.
4. When you define the model at page 3 left column and you introduce the λ_j as a generic function of the size, I would add that in all your experiments you use a particular case $\lambda^* = \log(|e_j|)$. This helps getting intuition. Perhaps motivate why you use this function in all your experiments.
5. Page 6, right column. Reference to Fig. 2 (c) should be to Fig. 2 (b) instead (Peaks), I believe.

6. Fig. 1: the meaning of the tables/matrices is unclear. They seem to show all the possible configurations of T , but this should be clarified. Otherwise I was left wondering if these were the adjacency matrices or similar, or the evolution $T_i(t)$. This latter quantity would be more interesting to show than all the possible configurations.
7. Fig. 1: Change $T_{1=0}$ to $T_1 = 0$, i.e. the = symbol should be on the T level, I believe. Same for all other instances of this in the same figure.
8. Fig. 3: Is the node ordering according to Branch I or III? In the caption is written I but in the text is III.
9. Fig. 4: please add colorbar for a).

DETAILED REPLY TO THE REPORT OF REVIEWER 1

Comment 1

This manuscript explores the higher-order dynamics introduced in Ref. [10] and discover multistability, intermittency, and hybrid transitions in the real-world synthetic hypergraph. The authors investigate the origin of such phenomena and show that the phenomena can be at least partially explained by the community structure of the hypergraph. The manuscript shows that the strength of the connection between the communities, which is characterized by the number of bridges (even though bridge was not clearly defined in the manuscript, which is necessary before publication), is closely related to the phenomena and concludes with the implications of the results to various social systems: Fewer bridges create multistability, while the creation of bridges induces intermittency.

The paper tackles one of the most vibrant research topics in the field to present interesting and novel results. Methods are clearly illustrated and appropriate for the research. Thus, I deem the manuscript suitable for publication in Nature Communications. However, to gain my definitive approval, the author should address the issues listed below:

Reply 1.— We thank the reviewer for their thorough and positive evaluation of our work. We agree about the referee’s concerns regarding the definition of ”bridges,” which we better defined in this revised version. Indeed, we incorporated all the reviewer’s comments. We hope that the revised manuscript merits publication.

Comment 2

- “Bridge” seems to be an important quantity in the manuscript. Nevertheless, the quantity was not clearly defined. It may be good to define it in Fig.1 and related content.

Reply 2.— Thank you for pointing that out. We agree that a formal definition of the concept of Bridges is necessary. Bridges are hyperedges composed by nodes that belong to different communities.

Action taken 2.— We defined the concept of bridges in the introduction text as

We demonstrate that these features could be linked to the community structure in the hypergraph and we show that bridges between communities play a crucial role. Here, we define bridges as hyperedges that are composed by nodes belonging to different communities.

Moreover, in the presentation of our results, in Section “Example of real-world hypergraphs: the case of blues reviews,” we also defined it again to make the text easier to read. This text reads as

To better understand the localization properties of our process, we focus on the probability that an individual is active, sampled from the simulations. In Fig.3 (a) we present the (hyper-)adjacency matrix, as in Eq (1), while in Fig.3 (b) to (d) we show the individual probabilities extracted from branches I to III, respectively. The matrix is ordered according to the individual probabilities of Branch III (lower). This figure shows that Branch III (panel (d)) is constrained to a group of nodes (a community roughly defined as $C_1 = \{v_1, v_2, \dots, v_{600}\}$), while Branch II (intermediate branch, panel (c)) is restricted to a different set of nodes together with some bridge hyperedges, and branch I accounts for the activation of all the nodes (see panel (b)). Here, we recall that bridges are defined as the hyperedges that are composed by nodes in different communities.

Additionally, we changed Fig. 1 to a panel, including a graphical representation of the concepts of bridges. In this case, we also changed the caption of this figure to

Graphical example of the social contagion model on a hypergraph. In (a) we present the example of a hypergraph. The tables next to each hyperedge and with the same color represent all the possible microstate configurations and its respectively associated group variable T_j . In (b) we show the graphical representation of one exemplary instance. In this representation, the black crosses represent the deactivation processes, N_i^ϕ , $1 \rightarrow 0$. For node i , the dashed lines represent inactive nodes, $Y_i = 0$, while continuous lines represent active nodes, $Y_i = 1$. In this specific example, the critical-mass threshold is $\Theta^* = 0.5$, the initial conditions are $Y_1 = Y_4 = Y_5 = 1$ and $Y_2 = Y_3 = Y_6 = Y_7 = 0$, and the red and blue crosses mark the time that the processes $N_j^{\lambda_j}$ activate all the inactive nodes in e_2 and e_3 , respectively. Moreover, on the right side of (b), we show the temporal evolution of the T_j variables in our exemplary instance. In this case, dashed lines indicate when the T_j 's are equal to zero, each movement to the left represents an increase in T_j , while each movement to the right indicates a decrease. In (c) we show a graphical example of the concept of bridges for two communities. Bridges are hyperedges that connect two communities, or groups of densely connected nodes.

Comment 3

- The methods used to determine which branch a particular simulation belongs to should be explained. If I understand correctly, two modes are considered to be in separate branches if there is zero probability of transition between them. It should be emphasized that transitions between the branches do not occur, or at least never occurred in the numerical simulations in this research.

Reply 3.— Thank you for pointing that out. Our model strongly depends on the initial micro-state, which might generate bimodal distributions or multiple stable branches for the same parameters. In the bimodal distribution case, we have an intermittent temporal behavior, and the main challenge here is sampling for long enough. On the other hand, when the probability of moving from one branch to the other is very low (zero in the thermodynamic limit and not found in our numerical simulations), we have multiple stable branches for the same value of parameters. In this case, the difficulty is finding the initial condition that will lead to such a solution.

Action taken 3.— We improved the text of our “Multistability and simulation methods” section, Sec. II.C.3 in our previous version and now Sec. IV.D., at the end of the manuscript due to the journal’s format.

Comment 4

- Caption of Fig. 3: The caption states the adjacency matrix is ordered according to the activity of Branch I. Panels (b), (c), and (d) seem to be ordered according to the activity of Branch III. The main text states that the matrix is ordered according to the individual probabilities of Branch III. The orderings of the nodes in the figure need to be clarified.

Reply 4.— Thank you for pointing that out. This was a typo. Indeed, the nodes are sorted according to the activity in Branch III.

Action taken 4.— We corrected the caption of Fig. 3, which now reads

Structure and dynamics of the blues reviews hypergraph. In (a) we plot the adjacency matrix of the blues reviews hypergraph ordered according to the activity of Branch III. In (b) to (d) we show the probability of being active in branches I to III respectively. The probabilities were estimated using Monte Carlo simulations with $\lambda = 0.1016$, $\delta = 1.0$, $\lambda^* = \log_2(|e_j|)$ and $\Theta^* = 0.5$.

Comment 5

- The word “experiment” in the manuscript can be misleading. It should be replaced with “numerical simulation” or another equivalent word that accurately describes the work of this research.

Reply 5.— Thank you for pointing that out. Although the use of the word “experiment” in the field of networks and complexity is employed in similar contexts, we do agree that substituting it with “numerical simulation” or equivalent words improves our text.

Action taken 5.— We changed the the word “experiment” by “numerical simulations” or simply “simulations,” depending on the context. We highlight that, we left the word “experiment” as is when referring to real-world experiments in controlled environments, similar to the ones in Ref. [L1].

Comment 6

- Page 3: This happens at the first time of the Poisson processes, which means that as soon as the threshold is hit, the inactive vertices become activated instantly after a random time exponentially distributed with parameter λ_i .”

=> It is not clear what the phrase “first time of the Poisson processes” means.

=> “as soon as” and “instantly” are contradictory to “after a random time exponentially distributed”. If I understand correctly, the word “instantly” should be replaced with “simultaneously”. Moreover, the contagion event does not happen as soon as the threshold is hit. Maybe only the time of the contagion event is determined in the numerical simulation algorithms as soon as the threshold is hit. It should be clarified.

Reply 6.— Thank you for pointing that out. We do agree that our previous formulation could have been more precise. The Poisson processes define the time scale of our process. When a hyperedge e_j reaches its internal threshold T_j , a Poisson process is created, effectively generating the random time for the associated event. As a consequence of the Poisson process, this time follows an exponential distribution with parameter λ_j . So, after this time passes, all the nodes in the hyperedge e_j change their state, becoming active. To better convey this message, we changed the description of our model.

Action taken 6.— We changed the sentence that described the group dynamics. It now reads as

So, given an hyperedge e_j , after the threshold is hit ($T_j \geq \Theta_j$) and a random time exponentially distributed with parameter λ_j passes (as a consequence of the Poisson process $N_j^{\lambda_j}$), all the the inactive vertices become activate simultaneously.

Comment 7

- Page 6: “the “frequency” at which the system switches between the two modes changes the variance and, therefore, the susceptibility.” => the relative time the system spends in each mode, rather than how often the transitions between the modes occur, determines the variance.

Reply 7.— Thank you very much for pointing that out. We agree with the reviewer’s suggestion.

Action taken 7.— We changed the sentence mentioned above. It now reads

In other words, the relative time the system spends in each mode changes the variance and, therefore, the susceptibility.

Comment 8

- On page 14: A very strong conclusion has been drawn with very little basis for the “pro-woman” laws–example. Either a stronger basis has to be presented, the claim has to be softened, or the example has to be removed from the manuscript.

Reply 8.— Thank you for pointing that out. We want to clarify that this discussion was introduced as speculation and was meant to propose/motivate future research lines in other fields. Our ration was to suggest that our observed group dynamics could provide new insights into this type of dynamics too. However, we agree that, in this specific example, many factors might also play an important role, and further studies should be performed in order. So, we decided to remove the discussion mentioned above.

Action taken 8.— We removed the discussion mentioned above.

Comment 9

A list of minor comments also follows:

- In the caption of figure 2, “In (a), (b) and (c), we present the order parameter, susceptibility, and the peaks” should be reordered as “In (a), (b) and (c), we present the order parameter, the peaks, and the susceptibility”

Reply 9.— Thank you for pointing that out. We corrected this issue in the text.

Action taken 9.— We changed the above-mentioned sentence in the caption. Now it reads as

In (a), (b) and (c), we present the order parameter, the peaks of the state distributions, and the susceptibility, respectively.

Comment 10

- On page 2, the indicator function needs to be defined, because it is not a commonly used symbol in the research community.

Reply 10.— Thank you for pointing that out. We agree that it is better to properly define the indicator function in the main text, as it will avoid confusion and help the reader from different areas.

Action taken 10.— We defined the indicator function after its first use after the transitions and rates definition. This explanation reads as

Current state:	Transition:	Rate:
each active v_i in \mathcal{V}	$1 \rightarrow 0$	δ ,
all inactive v_k in e_j	$0 \rightarrow 1$	$\lambda_j \mathbb{1}_{\{T_j \geq \Theta_j\}}$

where $\mathbb{1}_{\{\text{condition}\}}$ is the indicator function, which is one if the “condition” is satisfied, and zero otherwise.

Comment 11

- On page 3, the symbol for the ceiling function should be introduced in the text, because it is not a commonly used symbol in the research community.

Reply 11.— Thank you for pointing that out. We agree with the reviewer. As the journal is interdisciplinary, it is essential to define this function to make the message easier for a wider public.

Action taken 11.— We included the definition of the ceiling function after its first appearance as

It is also convenient to define $\Theta_j = \lceil \Theta^* |e_j| \rceil$, where $\lceil x \rceil$ is the ceiling function, which returns the least integer greater than or equal to x and Θ^* is a global parameter that is invariant to the cardinality of the hyperedges and lies in the range $0 \leq \Theta^* \leq 1$.

Comment 12

- In eq. (2), the condition on Y_i is written first in the indicator function, while in the text, the condition on T_j is written first. They should be equalized.

Reply 12.— Thank you very much for pointing that out. We should have noticed this in our first version. We corrected it.

Action taken 12.— We corrected the text, which now reads as

Furthermore, $\mathbb{1}_{\{Y_i=0, T_j \geq \Theta_j\}}$ is an indicator function that is 1 if $Y_i = 0$ and the critical-mass in the hyperedge is reached, and 0 otherwise.

Comment 13

- It is more appropriate that λ^* is introduced in Sec. II A rather than Sec. II B.

Reply 13.— Thank you for pointing that out. We agree with the reviewer and made the suggested modifications. Moreover, we should mention that, following Reviewer 3 suggestions, we also moved Sec. II B to the Methods section. Thus, making it necessary to define λ^* in Sec. II A. Also, we included a short justification for adopting $\lambda^*(|e_j|) = \log_2(|e_j|)$.

Action taken 13.— We added the following sentence to the last paragraph:

Moreover, we assumed that the spreading rate is composed by the product of a free parameter and a function of the cardinality, i.e., $\lambda_j = \lambda \times \lambda^*(|e_j|)$. In all of our numerical simulations we assumed $\lambda^*(|e_j|) = \log_2(|e_j|)$. This definition is convenient as, in the pairwise case, $\lambda^*(2) = 1$, guaranteeing that our dynamics reduces to the standard SIS model in a graph. Also, we choose $\log_2(|e_j|)$ as it grows sublinearly. So, in the limit of a large hyperedge, the average spreading rate tends to zero, i.e., $\lim_{|e_j| \rightarrow \infty} \frac{\log_2(|e_j|)}{|e_j|} = 0$.

Comment 14

- On page 4, “a random element of the list” is better replaced with “a random state in the list”

Reply 14.— Thank you for pointing that out. We agree that substituting the word “element” with the word “state” improves the precision and might avoid confusion.

Action taken 14.— The edited text now reads as

If the absorbing state is reached, then a random state in the list replaces the absorbing state.

Comment 15

- In figure 6 (a) and (d), it is impossible to distinguish where the blue dots are overlapped by red dots unless the figure is highly magnified. Because distinguishing it is important to interpret the figure, this problem should be resolved (Maybe by introducing transparency to the dots).

Reply 15.— Thank you for pointing that out. We agree that it was difficult to distinguish between the curves.

Action taken 15.— We changed Figure 6, introducing transparency to the dots and making the visualization easier at the superpositions.

Comment 16

- On page 10, “these transitions are discontinuous” → these transitions can be discontinuous

Reply 16.— Thank you for pointing that out. This suggestion makes the text more precise.

Action taken 16.— We changed the text to

However, when analyzing higher-order models, these transitions can be discontinuous [10, 12, 14].

Comment 17

- In the caption of figure 9, “here, in this figure” → “here,” or “in this figure,”

Reply 17.— Thank you for pointing that out. We corrected the caption.

Action taken 17.— We changed the first sentence of the caption of this figure to

For a “low” λ^* (in this figure \$\lambda^* = 10\$) we have a second-order phase transition followed by a hybrid transition.

Comment 18

- On page 10, it should be stated that the edges form a random regular network in the hyperblob.

Reply 18.— Thank you for pointing that out. This suggestion makes the text more precise.

Action taken 18.— We changed the definition of the hyperblob to

This hypergraph is defined as a set of homogeneous pairwise relationships, forming a random regular graph, where every node has $\langle k \rangle$ edges, together with a hyperedge containing all the nodes.

Comment 19

- On page 10, “As the nodes are indistinguishable” → Nodes are distinguishable in random regular networks.

Reply 19.— Thank you for this comment. The reviewer is correct. The nodes in a random regular graph are distinguishable. However, their degree is not distinguishable, as every node has the same degree. We agree that the word “indistinguishable” alone might be misleading. A more precise formulation would be using the term “indistinguishable by their degree,” in which we specify the “distinguishability” criteria. Note that, in our exact formulation, we assume that every node can effectively contact any other node, i.e., an annealed approach. However, in our simulations, we use a quenched approach, where we fix the structure. In the revision process, we realized that it would be positive to share some additional numerical simulations regarding the accuracy of this approximation. However, as this evaluation is not essential for the understanding of our paper, we opted to write a section in the SI.

Action taken 19.— We edited the text by changing the word “indistinguishable” to the term “indistinguishable by their degree” and added a short explanation about the meaning of this term. The new text reads as

As the nodes are indistinguishable by their degree we can describe the state of our system by the number of active nodes n .

In addition to the changes in the text, and as this approximation can raise questions about its accuracy, we wrote section III in the SI, showing and discussing the effects of quenched and annealed approaches.

DETAILED REPLY TO THE REPORT OF REVIEWER 2

Comment 1

This paper studies the dynamics of critical mass in collections of interconnected groups.

Reply 1.— First of all, we thank the reviewer for assessing our work.

Since we are unsure if we explained our model with sufficient detail, for the sake of precision, we would like to reiterate and clarify that our work studies the dynamics of a collection of critical-mass processes (also called threshold models). In our case, individuals might belong to more than one critical-mass process, this fact being one of the main ingredients that generate the novel observed behavior. We want to clarify that this is different from a single critical-mass process defined on top of interconnected groups. Conceptually the difference between the two systems is similar to a dynamics driven by the summation of a function, i.e. $\sum_{j \in \mathcal{E}_i} f(\sum_{i \in e_j} x_i)$ (our case), versus a system guided by the function of a summation, i.e. $f(\sum_{j \in \mathcal{E}_i} \sum_{i \in e_j} x_i)$ (the second case).

Comment 2

1. The authors adopt a computational (simulation-based) approach to the study of critical mass, and suggest that their approach to studying inter-group dynamics is novel. However, unbeknown to the authors, there is an enormous literature in social psychology, political science, and other fields that examine intergroup dynamics in large scale behavior change, starting with Allport's (1954) contact hypothesis, and extending through several decades of literature challenging it and extending it.

Reply 2.— We thank the reviewer for pointing us toward Allport's (1954) contact hypothesis. However, our model does not incorporate Allport's contact hypothesis explicitly. Thus, we left it as a possible future direction.

We want to emphasize that we are not suggesting that our approach to the problem is novel. On the contrary, our model is grounded on the literature of critical-mass models that span from social sciences [L1–L5] to theoretical works in physics [L6–L21]. Our model, first presented in [L17], assumes that the critical-mass models are correct, but extends them to the case in which individuals interact in multiple groups, rather than just one. Moreover, we also assume that interactions are always additive (see our answer to comment 3). Summarising, our contribution is the observation, classification, and analysis of the mechanisms that generate multistability, intermittency, and hybrid phase transitions.

Admittedly, our knowledge of the social sciences literature is limited compared to domain specialists. However, the interdisciplinary nature of our work and our results aim at reducing the gap between social sciences and mathematical models. Please see the last paragraph of our "Conclusion and perspectives" section, where we mention possible novel research lines that might reduce this gap.

In this spirit, we based our models on the well-accepted model of critical mass (also called threshold models) [L3] and followed a stochastic process approach that is well grounded in the literature of complex systems [L17–L19, L22, L23]. With this approach, our goal is to have a model that incorporates well-established concepts in social sciences within an elegant and convenient mathematical formulation. From this model, we could extract both theoretical results and hopefully motivate new research in social sciences, closing this interdisciplinary loop.

Action taken 2.— We included the following sentence in the third paragraph of the "conclusions and perspective" section.

Another foreseeable future direction would be incorporating different mechanisms as variants of the original model. For instance, one might propose variations that solve some of the above-mentioned limitations, e.g., including backlash or cultural opposition. Another possibility would be a variant that explicitly considers Alport's contact hypothesis [48].

Comment 3

2. The authors should also be aware that inter-group dynamics within a society or population often involve political and cultural divides across groups. Thus, critical mass within one group does not necessarily translate into critical mass within other groups, and may instead result in backlash, or cultural opposition from other groups.

Reply 3.— Thank you for pointing that out. Indeed our model does not take into account backlash or cultural opposition. But, again, this is a limitation of our current approach. However, we would like to remark that, in the case of disease spreading, as suggested by [L3, L18], our model would fit as is. Next, from a mathematical point of view, this is the most straightforward model that couples different critical-mass processes. We do understand that this is a simplification of reality. However, before understanding more complex behaviors (e.g., cultural opposition as suggested by the Reviewer), we need to build a theory for the most straightforward cases. For the sake of the exercise, let us suppose that a model with cultural opposition presented multistability. Then, the question would be, is multistability an effect of higher-order interactions, community structure, or cultural opposition? So, as we seek understanding, we have to begin with the simplest possible model that still has the features we are interested in (higher-order interactions). We are especially grateful for this comment as it points us to interesting new research lines.

Action taken 3.— We wrote a sentence about this limitation in our discussion section.

Another limitation we identified is that our model does not incorporate backlash or cultural opposition, which is important from a sociological point of view. Indeed, we assume that the activation of a group increases the probability of activation of other groups. However, this might only be the case in some real scenarios. Such extension is left as a future work.

We included the following sentence in the third paragraph of the “conclusions and perspective” section

Another foreseeable future direction would be incorporating different mechanisms as variants of the original model. For instance, one might propose variations that solve some of the above-mentioned limitations, e.g., including backlash or cultural opposition. Another possibility would be a variant that explicitly considers Alport's contact hypothesis [48].

Comment 4

3. I believe the authors are aware of the differences between empirical studies and simulation-based studies, however they note in the introduction to their article, that their model, “provides a theoretical foundation for, and phenomenological explanation to, the seemingly different experimental findings of expected critical mass thresholds. Studies based on a single group suggest a threshold between 30% – 40%. Conversely, a critical mass of 10% would correspond to a population composed of groups of diverse sizes, each one with a different threshold.” This is a problematic motivation for their study since it incorrectly characterizes the literature. The 10% result (Xie et al. 2011) refers to a simulation model based on the naming game, in which one population is overtaken by multiple ‘memes’ competing for dominance. Experimental results (Centola et al. 2018) show a critical mass of 25% for

empirical populations engaged in critical mass dynamics, and provide a mathematical model of these dynamics. And, ethnographic studies (Kanter 1977) show a critical mass of around 25% – 30%, but without analytical data. The second and third studies are both empirical and largely consistent with one another, while the first is a simulation with no empirical foundation. I do not see that there is a tension in the literature here, particularly since the present study is a simulation model without empirical grounding, so it is unclear how it will add to the current empirically grounded literature.

Reply 4.— Thank you very much for your comments and for pointing that out. Your comment inspired us to improve our introduction, including more recent literature. We admit that the sentence should have been more precise. We rewrote it. The main aim of this sentence is to put some of our model’s findings into the context of literature, both empirical and theoretical, showing that different threshold levels can be present at different aggregation levels. Perhaps one of the disagreements between our view and the reviewer’s points is the class of processes we consider as motivation. To the best of our knowledge, ethnographic studies [L2] and experimental results [L1] are indeed in agreement about a critical mass of around 25% – 30%. However, in our model, we are assuming the possibility of other types of processes. For instance, recent literature suggests that similar findings for social movements [L24]. We want to mention that more data-driven linguistic studies about norm changes, both in English and Spanish, showed that a committed minority of about 0.3% can impose a change [L25]. In the latter case, the observed threshold is considerably lower than the 25% obtained experimentally. One of the common ingredients in these studies is that they possibly involve interactions between multiple groups, contrasting with the single group in controlled studies. Thus, in this context, we believe our model can provide new insights.

Before concluding, we want to highlight that, although Ref. [L6] is a theoretical paper, in their conclusion, the authors motivate the relevance of their findings by historical examples, as, quoting the authors, “the suffragette movement in the early 20th century and the rise of the American civil-rights movement that started shortly after the size of the African-American population crossed the 10% mark (see the summary section in Ref. [L6]). Despite that, we agree that this reference could be misinterpreted and perhaps weak in the context of our previous version. So, we rewrote the sentence suggested by the reviewer, substituting our 10% example, in Ref. [L6], by the linguistic norm changes, which observed a threshold as lower as 0.3% in English and Spanish [L25]. Thus, we hope to have strengthened our argument by providing a data-driven reference. Moreover, this allows us to conjecture that when we have a collection of groups, they can have a higher critical threshold, and still, they might generate a collective behavior that is similar to a process with a lower threshold, which our model phenomenologically explains.

Action taken 4.— We rewrote part of the first paragraph of the introduction as

This evidence ranges from theoretical models [6–9] and observational studies [1, 3, 4], to real experimental approaches [5]. Although these studies suggest that the critical-mass threshold might range between 10% and 40%, there is evidence that it can be low as 0.3% in linguistic norm changes in English and Spanish [22, 23] or even just a few of individuals that are not comparable with the size of the population under study [23, 24]. Despite this wide range of observed thresholds, the critical-mass paradigm provides a reasonable abstraction to analyze and understand real social systems.

Moreover, we rewrote the fragment the reviewer mentioned as

Regarding questions (2) and (3), the model provides a theoretical foundation for, and a phenomenological explanation to, the seemingly different experimental findings of expected critical-mass thresholds. More specifically, ethnographic studies show a critical mass around 25% ~ 30% [1] and align with experimental results, which report a critical mass around 25% [5]. On the other hand, considering linguistic norm changes, the observed threshold is as low as 0.3% in English and Spanish [22]. The first studies consider a single group, while the linguistic norm changes consider a whole

population, which can be understood as a collection of groups. Thus, in the latter, we might have groups with different sizes, each with a different threshold. In other words, from the perspective of the model in Ref. [12], it is possible to have individual groups with thresholds between 25% and 40%, and, at the same time, due to the group intersections, having a critical mass at the population level around a much lower value.

Comment 5

4. The authors present their work as “a dynamical analysis of the social contagion model presented in Ref.[10].” There is an enormous variety of social contagion models, many of which have solid empirical foundations. I cannot understand why this article selects the model in Ref.[10] as the basis for its analysis. It does not seem to have much grounding or connection to the broader literature on contagion, but is a quite narrow model, which this paper then elaborates in more detail. This kind of modeling exercise does not seem suitable for a general interest scientific journal. Consistent with this assessment, I find the analysis of the ‘Blues Reviews’ hypergraph completely un-compelling since this seems like a fitting exercise, and it provides no substantive or micro-foundational model that would motivate the applicability of their particular theoretical approach to these data (nor would it explain why their theoretical approach would be expected, or not expected, to work for other kinds of data).

Reply 5.— We respect the reviewer’s opinion but strongly disagree with most of this comment’s points. We hope that by clarifying some specific issues, we can convince the reviewer otherwise. Below we provide details about these specific points.

1. First, we disagree with the sentence “It does not seem to have much grounding or connection to the broader literature on contagion.” Our model is a collection of critical-mass dynamics where individuals can belong to more than one group. Thus, we are only assuming that: (i) the critical-mass process is a reasonable model for the dynamics inside each group, justified by a vast literature in many fields, including social sciences [L1–L5] and theoretical analysis [L6–L21]; (ii) individuals can belong to multiple groups, and (iii) individuals have a binary state (active or inactive), which is a reasonable approach also used both in social sciences [L3] and in mathematical modeling works [L17–L19, L22, L23]. Moreover, our model can also be interpreted as a generalization of the SIS process to hypergraphs. We remark that every graph is a hypergraph with $\max(|e_j|) = 2$ (only pairs of nodes), and, in this case, the critical-mass process reduces to a standard spreading process. So, from the application on epidemic modeling, our model is also well grounded.
2. We also disagree when classifying our model as “narrow.” From the literature of social sciences, in [L3], the author argues that threshold models can be helpful to model: (1) diffusion of innovation, (2) rumors and disease, (3) strikes, (4) voting, (5) education attainment, (6) leaving social occasions, (7) migration, and (8) experimental social psychology. As our model is inspired by a particular extension of threshold models (where we consider a collection of threshold – also called critical-mass – models), one might think about the same applications. At this point, we must remark that we are not mentioning these applications in the main text as our work is mainly theoretical, and we feel that further studies are necessary for this direction. Moreover, our model is mathematically very flexible, allowing for many different formulations. As mentioned above, it extends the SIS process to hypergraphs. Dynamically, we obtained a very rich and unexpected phenomenology. Indeed, in our discussion and conclusion sections, we leave many open questions in different areas that we believe can be explored in the future. Finally, we remark that we still believe that there is still new phenomenology or new mechanisms that generate a similar phenomenology to be discovered in this model.

3. We do not understand how the analysis of the ‘Blues Reviews’ hypergraph can be interpreted as a fitting exercise. This hypergraph is only an example of how real-world systems, with correlations and mesoscale organizations, can present a very unexpected behavior. Moreover, there is no fitting in our paper. To design a fitting exercise, we would need dynamical data, which, to the best of our knowledge, is lacking in the literature (at least with the granularity and the temporal scale required by our analysis). We remark that the Blues Reviews hypergraph is a real system that presents a very rich dynamical repertoire. This served as a motivation for the development of our theory. Additionally, artificial systems explained all the observed behaviors in detail. This study could explain the mechanisms that generated the observed phenomenology. Interestingly enough, most of the observed behavior was linked to the existence of communities in the hypergraph. Finally, the organization in communities is indeed a ubiquitous feature in social systems in general.

In summary, we hope to have convinced the reviewer that our work is not ungrounded, narrow, or a fitting exercise. Most importantly, we hope that it is now clear that we aim to provide a theoretical contribution based on social sciences concepts in the spirit of interdisciplinary research. However, our primary focus is theoretical, and further studies in social sciences are necessary to bridge the gap entirely. This is partially expressed in our conclusion, where we suggest that our results might also help in the design of new controlled experiments.

Comment 6

5. The relationship between bridges, or in other words, “connectivity”, and multistable states is well-known, as it is essential for coordination across communities that those communities are sufficiently connected.

Reply 6.— Thank you for pointing that out. This is probably our fault. As Reviewer 1 pointed out, in the first version of our manuscript, we did not properly define the concept of “bridges.” We believe this might have generated some confusion. Please see also Comment 2 to Reviewer 1. We want to clarify that bridges are not a synonym for “connectivity.” In our context, bridges are hyperedges composed of nodes belonging to different communities. Since each hyperedge also represents a critical-mass process, this difference is fundamental, as it implicitly includes the notion of activity. Note that, while there could be a structure connecting two communities (that is, a bridge hyperedge), if it is not active (i.e., the critical-mass threshold was not reached), then the communities are effectively disconnected dynamically.

Moreover, we want to clarify the type of processes analyzed. Our manuscript studies a model that falls in the spreading process class. Therefore, we are not evaluating coordination dynamics directly, i.e., games as in Refs. [L16], nor synchronization processes, as in Ref. [L26]. As we are focusing on a “spreading-like” type of process, we want to recall that, when defined in a graph (which is a hypergraph with $\max(|e_j|) = 2$), our dynamics reduce to an SIS process.

Also, we would like to mention that we understand the statement that “multistable states is well-known,” however, to the best of our knowledge, this is not a common behavior in graphs. Most graph-based spreading models present second-order phase transitions with a single solution. Please see Refs. [L22, L23], for examples. For the sake of precision, we remark that, in our context, multistability means that, for a set of control parameters $(\Theta^*, \lambda, \delta)$, we have multiple possible order parameters (ρ , the average of active individuals). We also would like to highlight that multistability should not be confused with the coexistence of states, where we have regions of the parameter space in which multiple states coexist, but the average (over time) order parameter is unique. To conclude our argument about the novelty of our results, we mention another example of multistability in higher-order systems. The pre-print [L27] (submitted after our manuscript) also reported multistability in synchronization dynamics on a higher-order system. In this work, the authors also relate multistability with higher-order interactions in a community structured population, suggesting our original observation is probably a common phenomena across

different higher-order dynamical processes, and not specific to spreading dynamics. While this is still a conjecture at this stage, we believe that the additional evidence from [L27] supports our claim that our observations are novel and interesting.

Finally, we want to remark that our paper is not only about multistability in itself. Importantly, we also show how multistability and intermittency can be linked via the community structure in the hypergraph, providing a mechanistic explanation of the observed phenomena. Furthermore, we showed that it is possible to observe multiple transitions between branches. Finally, we provided additional insights about the type of transitions we observed (that is, hybrid ones).

Action taken 6.— Here we reproduce comment 2 to reviewer 1 answer. We defined the concept of bridges in the introduction text as

We demonstrate that these features could be linked to the community structure in the hypergraph and we show that bridges between communities play a crucial role. Here, we define bridges as hyperedges that are composed by nodes belonging to different communities.

Moreover, in the presentation of our results, in Section “Example of real-world hypergraphs: the case of blues reviews,” we also defined it again to make the text easier to read. This text reads as

To better understand the localization properties of our process, we focus on the probability that an individual is active, sampled from the simulations. In Fig. 3 (a) we present the (hyper-)adjacency matrix, as in Eq. (1), while in Fig. 3 (b) to (d) we show the individual probabilities extracted from branches I to III, respectively. The matrix is ordered according to the individual probabilities of Branch III (lower). This figure shows that Branch III (panel (d)) is constrained to a group of nodes (a community roughly defined as $C_1 = \{v_1, v_2, \dots, v_{600}\}$), while Branch II (intermediate branch, panel (c)) is restricted to a different set of nodes together with some bridge hyperedges, and branch I accounts for the activation of all the nodes (see panel (b)). Here, we recall that bridges are defined as the hyperedges that are composed by nodes in different communities.

Additionally, we changed Fig. 1 to a panel, including a graphical representation of the concepts of bridges. In this case, we also changed the caption of this figure to

Graphical example of the social contagion model on a hypergraph. In (a) we present the example of a hypergraph. The tables next to each hyperedge and with the same color represent all the possible microstate configurations and its respectively associated group variable \$T_j\$. In (b) we show the graphical representation of one exemplary instance. In this representation, the black crosses represent the deactivation processes, $N_i^0, 1 \rightarrow 0$. For node i , the dashed lines represent inactive nodes, $Y_i = 0$, while continuous lines represent active nodes, $Y_i = 1$. In this specific example, the critical-mass threshold is $\Theta^* = 0.5$, the initial conditions are $Y_1 = Y_4 = Y_5 = 1$ and $Y_2 = Y_3 = Y_6 = Y_7 = 0$, and the red and blue crosses mark the time at which the processes $N_j^{\lambda_j}$ activate all the inactive nodes in e_2 and e_3 , respectively. Moreover, on the right side of (b), we show the temporal evolution of the \$T_j\$ variables in our exemplary instance. In (c) we show a graphical example of the concept of bridges for two communities. Bridges are hyperedges that connect two communities, or groups of densely connected nodes.

Comment 7

6. The analyses provided in this study vary between i) confirming existing theoretical intuitions about dynamics

of coordination in graphs, and ii) being so specific to the hypergraph-theoretic architecture upon which this study is based that the counterintuitive results are not convincing in light of the article’s claims of generality.

Reply 7.— We strongly disagree with this comment. We hope our answers to the previous comments clarify that our work can not be reduced to confirming the results in the literature. In summary, our main results show the existence and mechanisms that generate multistability, intermittency, and hybrid phase transitions in the social contagion model in hypergraphs. Our model is in the class of spreading processes.

Our claims of generality are supported not only by our work but also by both previous work by us and on-going work by other researchers. For instance, the pre-print [L27] essentially confirms our conjecture about the generality of multistability in hypergraphs with community structure. Quoting a paragraph in our previous “conclusions and perspective section:

“We hope our results open new paths for exploring social contagion models in hypergraphs. Analytically, understanding the necessary and sufficient conditions for observed phenomenology is one of the most challenging future problems. From a numerical perspective, the exploration and characterization of other real systems might also reveal so far unobserved behaviors as well as confirm our findings. Another view would be motivating further research about understanding the impact of our results on different processes. For instance, how can localization impact on synchronization of oscillators? Would we have multistability in such dynamics?”

Moreover, concerning hybrid phase transitions, our conjecture about its generality is justified by the fact that it was already observed in simplicial complexes with power-law degree distribution. Our result is the analytical demonstration (exact formulation) that hybrid phase transitions are more general and do not depend on simplicial complexes or power-law degree distributions.

To conclude our arguments about the theoretical generality of our model, we want to remark that our model is well-formulated and also naturally accommodates graph models. Indeed, every graph is also a –very simple– hypergraph. Therefore, “the hypergraph-theoretic architecture” is at least as general as any “graph-theoretic” model. As already extensively mentioned in this rebuttal and in the main text, our model reduces to an SIS model if we consider a graph –instead of a hypergraph– as substrate for the process. Finally, by considering a simplicial complex as the hypergraph and using $\Theta_j = |e_j| - 1$, we recover the simplicial contagion model from Ref. [L10]. We must also recall that many real-world systems are built with groups (e.g., WhatsApp or Telegram) as their building blocks, and, in the case of these systems, the graph is a simplification. Sometimes this simplification is justified. However, in many other cases, it might not be. Here we point to a class of problems in which this simplification would fail as an SIS on a projected graph will not present the same phenomenology: it will present second-order –continuous– transitions instead of hybrid phase transitions, no multistability not intermittency.

Next, regarding the applications, we already mentioned a few concrete research lines in our previous conclusion. Note that we included lines ranging from controlled experiments in social sciences to data science. Here, our clear goal is to identify the strengths and weaknesses of our models. Quoting our previous conclusion:

“Our findings might also impact the design of real experiments similar to the ones in Refs. [L1, L28, L29]. One of the main difficulties with this type of experiment is that the number of people participating is often reduced, and the signals in the observables are usually noisy. In such small systems, while accurately measuring multistability might be challenging, intermittency might be easier to capture as we would be interested in finding periods of high activity followed by periods of low activity. Along similar lines, data coming from online social systems, while abundant in volume and number of potential subjects, is less controlled, imposing limitations on the modeling possibilities.

Despite these limitations, there are still many available datasets that are higher-order in nature (i.e., the most natural representation would be a group and not a collection of pairwise interactions), for instance, WhatsApp message exchange in groups (see Ref. [L30, L31]) or data from Reddit as the collaboration in the social experiment r/place [L32]. We remark that, in principle, studying these datasets from the viewpoint of higher-order interactions is possible.”

Despite these concrete examples, we rewrote the last paragraph of our “conclusion and perspectives” section, extending our possible range of applications using the catalog provided in [L3]. Also, we better justified the epidemic spreading cases using recent literature, Ref. [L18]. Finally, aiming at a wider public, we conjectured a few examples in the book “Tipping point: How Little Things Can Make a Big Difference” [L33], by Malcolm Gladwell, better justifying the fashion example that was already present in our previous conclusion. With these examples, we hope we convinced the reviewer that our findings are neither restricted to confirming the theoretical results nor hypergraph-specific findings.

Action taken 7.— In addition to the changes in the introduction mentioned in the comments above, we rewrote the last paragraph of our conclusion as

To conclude, the literature on threshold models suggests that many processes can be modeled as binary choice critical-mass processes. For example, in Ref. [2], the author proposes a catalog of processes that includes diffusion of innovation, rumors and diseases, strikes, voting, educational attainment, leaving social occasions, migration, and experimental psychology. We must highlight that in Ref. [2], the author associates the threshold processes to the individuals and not the groups. However, the threshold is reached or not due to individual social interactions. Our approach is slightly different as we focus on the group rather than the individuals. Despite these differences, the proposed catalog is still valid in our case. The main difference is that our model might provide different mechanistic explanations for similar phenomena. We should also complement the argument for the case of disease spreading following a similar reasoning as in [18]. We presume that our model may provide new insights into a disease spreading in which there is a viral load threshold [49]. Since, in this case, sharing an environment with a few infected people might impose an increased risk higher than linear, which would be the standard complex network prediction, our model could better explain this process. Finally, we could also mention examples from our daily lives that can be conjectured as a result of group interactions. For example, some of the phenomena described by Malcolm Gladwell in his book, “Tipping point: How Little Things Can Make a Big Difference” [50] can also be interpreted or re-analyzed from the group dynamics point of view. A notable example would be the famous saying that “fashion is cyclic” is an effect of group interactions as fashion can be understood as a norm, as in [5,23]. In this scenario, we hypothesize that the observed cyclic behavior is associated with the structural organization of our societies.

DETAILED REPLY TO THE REPORT OF REVIEWER 3

Comment 1

The authors study the behavior of a social contagion model that they previously proposed. They find a rich set of behaviors that is different than that usually observed in standard pairwise graphs and they associate it with community structure. The results are interesting, in particular the rich phenomenology of intermittency and metastability. The paper is clear and well written. I think the paper makes a relevant contribution and deserves publication. I have few comments for suggestions to make it even stronger, as I describe below.

Reply 1.— We thank the reviewer for their constructive evaluation of our work. Indeed, we incorporated all the reviewer’s suggestions, which, in our opinion, made the paper significantly stronger. We also would like to remark that, due to the reviewer’s comments, we could extend our knowledge about the role of bridge hyperedges in multistability and intermittency. Specifically, we found that, in addition to the number of bridge hyperedges, a similar interplay between multistability and intermittency can also be found by changing the size of the hyperedges or the critical-mass threshold.

Comment 2

Major points 1. The paper is quite long and dense of results. While there are several results that the authors would like to present, I think it is worth prioritizing and shift some of them into the SI for better highlighting the most important ones. For instance, I believe that the description of the hyperblob case in Sec V could be condensed. Many details of the calculations can go in the SI and they could compress that section by focusing on the main results instead.

Reply 2.— We thank the reviewer for the suggestion. We agree that many technical details and derivations should be moved to the methods section. This change should also make our results more evident and make it easier to read. Moreover, by doing that, we also comply with the journal’s format.

Action taken 2.— We created the “Methods” section at the end of the paper. To this section, we moved the following subsections that were previously in our main text:

- The first-order approximation (individual-based): previously section II.B;
- Continuous-time simulations: previously section II.C.1;
- Quasi-stationary method (QS): previously section II.C.2;
- Multistability and simulation methods: previously section II.C.3;
- Artificial hypergraph model: previously section IV.B;
- Exact equations for the hyperblob: previously section V.A;
- Quasi-stationary steady-state solutions: previously section V.A.1;

Comment 3

2. The real data they used is very sparse, $N = 1106$ and only 694 hyperedges. I am wondering if results would change for sparsity that is still sparse but higher than this quite low value. Maybe results would be less sensitive to initial condition? Maybe we see less branches? I would also recommend to report few more statistics about this datasets, e.g. the distribution of $|e_j|$, node degree distribution and something describing how many lower size hyperedges are already contained into bigger ones (e.g. how many pairs or triangles are contained into hyperedges of $|e_j| > 2, 3$, to measure their redundancy).

Reply 3.— Thank you very much for sharing these comments and suggestions with us. Indeed, in our discussions to provide a better example of the mechanisms generating the different branches, we decided to propose a new experiment that hopefully will help to understand the roots of multistability.

We propose a “recipe” to generate hypergraphs that will have as many transitions as one wishes (in the SI, we showed an example with 4). In this experiment, we also performed a parameter exploration in Θ^* , the critical-mass threshold. We showed that the number of communities is enough to determine the number of transitions, where the process remains active in different sub-groups of nodes with different sizes, in alignment with the results presented in the main text. Additionally, as the communities have different densities, we also have evidence that the main driver for the multistability, and consequently the number of branches, is related to the number of communities and not the density of the hypergraph itself. It is worth mentioning that, in our experimental setup, the difference in sparsity between communities is necessary.

Importantly, we are not claiming that this is the only mechanism that generates multistability but the one we found. It is possible that other mechanisms also generate a similar phenomenology. Indeed, this is partially the reason we are excited about our findings. We believe that they might also open new research opportunities, seeking different mechanisms and new phenomena.

Finally, we remark that we opted to propose to answer these comments using artificial hypergraphs for two reasons: (i) the experiment is better controlled, and we avoid dealing with eventual correlation effects, and (ii) despite being an interesting exercise, adding or removing hyperedges of a real system might be more difficult to justify.

Action taken 3.— Concerning the questions Maybe results would be less sensitive to initial condition? and Maybe we see less branches?, we wrote a section on the SI, “A four community hypergraph example,” where we show that we are able to control (up to some degree) the number of branches and that each branch is connected to a different initial condition.

Moreover, regarding the suggestions about reporting more statistics about the blues review dataset, we wrote a section on the SI, “Blues reviews: Structural characterization,” where we report the degree and cardinality distributions. In the same section, we also report a few metrics about the hyperedge interactions, providing additional insights about their redundancy.

Comment 4

3. Similarly, I would be interested to understand how results are impacted by the distribution of hyperedge sizes. For instance, one could check if we see similar behaviors when varying the max hyperedge size. I expect that when this is small, e.g. 3-5, we get a more similar picture to the pairwise case. So it would be interesting to see if there is a critical value after which, e.g., intermittency or multistability is observed. My intuition is that what makes the behavior rich and different from the pairwise graph case is the overlap between hyperedges. This becomes

more important as hyperedges become bigger. It may also depend on Θ^* . Hence this type of analysis should be investigated.

Reply 4.— Thank you for your suggestions. To shed light on the impact of the max hyperedge size, we focus on the same artificial experiment as in the main text (two communities), but varying μ , the parameter of the exponential distribution of cardinalities. As a consequence, we are evaluating structures with smaller hyperedges for lower values of μ and structures with bigger hyperedges for larger values of μ . We opted to use the model in the main paper as we believe that would be the most elegant solution and also to avoid any potential confusion.

From this experiment, we observed that, for a fixed number of bridge hyperedges, when the hyperedge sizes are small, the transition is abrupt, and we might also have multistability. On the other hand, when the bridge hyperedges are bigger, we observe intermittent behavior. At a mesoscale, we have a bimodal distribution of states, while at a macro scale, we have a peak in the susceptibility curve. Despite being caused by different mechanisms, this is the same behavior as observed in the main text, where we change the number of bridges but keep their size fixed (on average). This is indeed a new result, and we added a comment about this in the main text.

Furthermore, the notion of sparsity and the concept of bridges (both in a dynamical sense) also change as we also change the critical-mass threshold Θ^* .

Action taken 4.— Regarding the newly observed interplay between the size of the bridges and multistability and intermittency, we added the following sentences in the main text

These results suggest that when bridges are scarce, the communities are dynamically disconnected. Hence, we might have multiple stable solutions for a range of λ due to localization. As we add bridging hyperedges, we allow the process to travel across communities. However, this can destroy the multiple stable solutions by merging them into a bimodal distribution of states and creating intermittency. We highlight that a similar effect was also observed by increasing/decreasing the hyperedge cardinalities and by changing the critical-mass threshold Θ^* . In the first scenario, we noticed that by increasing the average hyperedge cardinality, we could change our system's behavior from multistability to intermittency. Particularly, by i) considering the same artificial model with communities as in previous numerical simulations, ii) fixing the number of hyperedges and bridges, but iii) changing the average hyperedge cardinality, μ , we were able to observe a shift from a multistable region for low μ to an intermittent behavior for larger μ . Moreover, by changing the critical-mass threshold Θ^* , we observed that, for higher values of Θ^* , we tend to favor multistability, while for lower values of Θ^* we favor intermittency. The numerical simulations of changing μ and Θ^* are presented in the SI. It is worth highlighting that it might be possible to construct more complex hypergraphs that would display more branches and possibly even allow for multistability and intermittency at the same region of λ . Please see also the SI for an example with four communities. We remark that here we focused on the simplest structure that reproduces both phenomena. Furthermore, one can see a relation between our results and the previous findings [L34] relative to the identification of network structures and of individuals best suited for spreading complex contagions. The authors proposed a centrality measure that accounts for the number of “enough wide bridges” between two nodes. Although in [L34] they are still using graphs (but the contagion is complex), this concept resembles the ideas behind critical-mass processes associated with our hyperedges. Thus, the term “enough wide bridges” might be understood as an abstraction of the critical-mass threshold in our context. We remark that the term “enough wide bridges” summarizes our results as it incorporates both the number of bridges (as shown in Fig. 6) and “how easy” it is to activate these bridges (results reported in the SI).

Complementary, these new results are reported in the SI in the section entitled “Additional analysis of the two community experiments.”

Finally, we included our exploration of Θ^* in the section “A four community hypergraph example” in the SI.

Comment 5

Minor points 1. How many rewiring did you consider in the randomly rewired version of Fig.2 ? I cannot see error bars to understand whether there were few instances where the rewiring could lead to behaviors similar to the real case.

Reply 5.— Thank you for your comment. We agree that this information is indeed missing in our previous version. We provided a better description of the method employed to randomize the real hypergraph as well as a link to the GitHub repository with the code used to generate such randomizations.

Moreover, to answer the question of whether some instances where the rewiring procedure could lead to a similar behavior as the real case, we performed 30 additional randomized hypergraphs and their respective simulations. We observed that all the simulations have similar behavior. The critical point might slightly change from one randomization to another. However, the discontinuous transition is always present. The difference between different curves is indeed very small, and the curves are visually indistinguishable, implying a low variance between simulations. Moreover, a small peak of susceptibility after the discontinuity was also found in all our experiments. Thus, one can conclude that the results reported in the main text are representative of random versions of the Blues Reviews hypergraph. Note that if we had performed an insufficient number of rewirings, we would also expect a reminiscent behavior from the real case. Finally, we remark that these results are reported in the SI.

In the main text, in Fig. 2, we kept only one simulation because the QS simulations as averaging over different hypergraphs might make the discontinuous transition look like a continuous one. Note that, as the discontinuity point is slightly different from one randomized version to another, thus, averaging the curves will smooth the transition and, therefore, pass the wrong message.

Action taken 5.— We incorporated additional information about the hypergraph configuration model, including also the number of rewirings in the main text. The text now reads as

Fig. 2 shows the QS Monte Carlo simulations (see the Methods section for more details about this method) for our social contagion model in the blues reviews hypergraph and in a randomly rewired version obtained from the exact version of the vertex-labeled hypergraph configuration model presented in [28] (Algorithm 2 in [28] and code from [29]) after 10^7 rewirings. Moreover, in the SI we present 30 additional Monte Carlo simulations for different randomizations of the blues reviews hypergraph, showing that they have a similar behavior.

Despite the description mentioned above, in the SI, we show 30 different executions of the QS method, where we show that the behavior reported in Fig. 2 in the main text is representative of a randomized version of the blues reviews hypergraph.

Comment 6

2. It is not clear if the results on the Blues reviews are obtained using only the giant connected component accounting for the hyperedges. They mention the pairwise graph giant component of 24 nodes, but is not clear if

accounting for hyperedges we obtain a unique component of 1106 nodes. Please clarify this.

Reply 6.— Thank you for pointing that out. We do agree that clear information about the size of the giant connected component is missing. We corrected this issue by changing the description of the Blues reviews hypergraph, clarifying that the pairwise giant connected component includes only 24 nodes, while the giant component for the whole hypergraph has 1106 nodes.

Action taken 6.— We clarified this point by rewriting the description of the real hypergraph, that now read as

This hypergraph has $N = 1106$ nodes and 694 hyperedges, whose maximum cardinality is $\max(|e_j|) = 83$. In this dataset, the pairwise interactions are sparse, which alone would form a giant component of only 24 nodes. However, by accounting all the hyperedges, the giant component of the hypergraph has $N = 1106$ nodes. We remark that repeated hyperedges were not allowed. Moreover, for a structural analysis of this hypergraph, we refer to the Supplementary Information (SI).

Comment 7

3. Is Eq 6 an approximation as written in the text? In this case I would add change the symbol = with the symbol \approx . As this equation comes from a reference it is hard to tell if it exact or it involves some kind of approximation.

Reply 7.— Thank you for pointing out this confusion. The probabilities in equations 5 and 6 (now equations 7 and 8) are exact if the probabilities are independent. So, the approximation considers that y_j 's are independent, which is expressed by Eq. 4 (now Eq. 6).

Action taken 7.— We added the following sentence at the end of the sub-section “The first-order approximation (individual-based)” in the methods section.

Note also that, Eq. (6) is an approximation as we assume that the nodes' state is independent. However, equations (7) and (8) are exact for independent random variables and are also identical, giving the same results.

Comment 8

4. When you define the model at page 3 left column and you introduce the λ_j as a generic function of the size, I would add that in all your experiments you use a particular case $\lambda^* = \log(|e_j|)$. This helps getting intuition. Perhaps motivate why you use this function in all your experiments.

Reply 8.— Thank you for this suggestion. The reason for such a definition is that we opted to define our model in its most general formulation. However, this indeed might create some confusion. Thus, we follow the reviewer's suggestion, defining $\lambda^* = \log(|e_j|)$.

Action taken 8.— We added the following sentence after the model definition

Moreover, we assumed that the spreading rate is composed by the product of a free parameter and a function of the cardinality, i.e., $\lambda_j = \lambda \times \lambda^*(|e_j|)$. In all of our numerical simulations we assumed $\lambda^*(|e_j|) = \log_2(|e_j|)$. This definition is convenient as, in the pairwise case, $\lambda^*(2) = 1$, guaranteeing that our dynamics reduces to the standard SIS model in a graph. Also, we choose $\log_2(|e_j|)$ as it grows

sublinearly. So, in the limit of a large hyperedge, the average spreading rate tends to zero, i.e., $\lim_{|e_j| \rightarrow \infty} \frac{\log_2(|e_j|)}{|e_j|} = 0$.

Comment 9

5. Page 6, right column. Reference to Fig. 2 (c) should be to Fig. 2 (b) instead (Peaks), I believe.

Reply 9.— We thank the reviewer for pointing that out. In this revised version, we found a couple of points where we mistakenly exchanged the references of Fig. 2 (b) and (c).

Action taken 9.— We corrected all the mistaken references. For the sake of space, we left these changes only on the marked manuscript.

Comment 10

6. Fig. 1: the meaning of the tables/matrices is unclear. They seem to show all the possible configurations of T , but this should be clarified. Otherwise I was left wondering if these were the adjacency matrices or similar, or the evolution $T_i(t)$. This latter quantity would be more interesting to show than all the possible configurations.

Reply 10.— Thank you for pointing that out. We decided to show the set of all possible configurations because they are closely related to the approximations in Eq. (6). In this equation, we need to account for all possible configurations to obtain the probability of activation of a given hyperedge. We believe that, despite being simple, this concept is essential to understand the process and how one can generalize spreading processes from graphs to hypergraphs. However, we agree that showing the temporal evolution of T_j might also help the reader to understand the process. So, we changed Fig. 1, including $T_j(t)$ for our example.

Action taken 10.— We changed Fig 1, including $T_j(t)$, and its caption, explaining the meaning of the mentioned tables. The caption now reads as

Graphical example of the social contagion model on a hypergraph. In (a) we present the example of a hypergraph. The tables next to each hyperedge and with the same color represent all the possible microstate configurations and its respectively associated group variable T_j . In (b) we show the graphical representation of one exemplary instance. In this representation, the black crosses represent the deactivation processes, $N_i^0, 1 \rightarrow 0$. For node i , the dashed lines represent inactive nodes, $Y_i = 0$, while continuous lines represent active nodes, $Y_i = 1$. In this specific example, the critical-mass threshold is $\Theta^* = 0.5$, the initial conditions are $Y_1 = Y_4 = Y_5 = 1$ and $Y_2 = Y_3 = Y_6 = Y_7 = 0$, and the red and blue crosses mark the time at which the processes $N_j^{\lambda_j}$ activate all the inactive nodes in e_2 and e_3 , respectively. Moreover, on the right side of (b), we show the temporal evolution of the T_j variables in our exemplary instance. In (c) we show a graphical example of the concept of bridges for two communities. Bridges are hyperedges that connect two communities, or groups of densely connected nodes.

In our model definition, section we explained the meaning of these tables as

See the tables next to each hyperedge in Fig. 1 (a) for a graphical representation of all the possible microstates and the T_j variables.

Furthermore, in the methods section, we referenced and commented on the above-mentioned tables by adding the following description after the equations 6 and 7

where F_ℓ is the set of all subsets of ℓ integers from $\{1, 2, \dots, n = |e_j|\}$, A is one of those sets, and A^c is its complementary. Intuitively, A accounts for the possibly active nodes and A^c the inactive ones. Thus, the summation over F_ℓ considers all possible micro configurations in a given hyperedge. In Fig. 1 (a) we show an example of the possible micro states for each hyperedge and their associated value of T_j .

Comment 11

7. Fig. 1: Change $T_1=0$ to $T_1 = 0$, i.e. the $=$ symbol should be on the T level, I believe. Same for all other instances of this in the same figure.

Reply 11.— Thank you very much for pointing that out. The reviewer is right.

Action taken 11.— We changed Figure 1 and corrected this issue.

Comment 12

8. Fig. 3: Is the node ordering according to Branch I or III? In the caption is written I but in the text is III.

Reply 12.— Thank you for pointing that out. The ordering follows the activity of branch III.

Action taken 12.— We corrected the caption of Fig. 3, which now reads as

Structure and dynamics of the blues reviews hypergraph. In (a) we plot the adjacency matrix of the blues reviews hypergraph ordered according to the activity of Branch III. In (b) to (d) we show the probability of being active in branches I to III respectively. The probabilities were estimated using Monte Carlo simulations with $\lambda = 0.1016$, $\delta = 1.0$, $\lambda^* = \log_2(|e_j|)$ and $\Theta^* = 0.5$.

Comment 13

9. Fig. 4: please add colorbar for a).

Reply 13.— Thank you for pointing out this confusion. In Fig. 4 (a), we only have two levels, active and inactive. Unfortunately, we can not have a vectorial image here as we would need $1106 \times 2.98 \times 10^5$ pixels, constraining ourselves to a png format. As a side effect, we have aliasing, which might generate this confusion. To solve this issue, we wrote a comment in the caption, clarifying that we only have two levels.

Action taken 13.— We modified the caption of Fig. 4, which now reads

Intermittent behavior in the blues reviews hypergraph. In (a) we show the graphical visualization of the activity of the nodes in a specific run of the MC. **Pixels in dark represent inactive nodes, while colored pixels represent active nodes. The intermediate colors in the figure are only an effect of aliasing and do not have a physical interpretation.** The nodes are sorted by their activity for visualization purposes. In (b), we represent the order parameter as a function of time for the same simulation. The dynamical parameters for this simulation are $\lambda = 0.1016$, $\delta = 1.0$, $\Theta^* = 0.5$, and $\lambda^*(|e|) = \log_2(|e_j|)$. Notice that, in (a) we depict the dynamics as a function of the iterations, while in (b) as a function of the time, explaining the small displacement between both figures.

-
- [L1] D. Centola, J. Becker, D. Brackbill, and A. Baronchelli, *Science* **360**, 1116 (2018).
- [L2] R. M. Kanter, *American Journal of Sociology* **82**, 965 (1977).
- [L3] M. Granovetter, *The American Journal of Sociology* **83**, 1420 (1978).
- [L4] D. Dahlerup, *Scandinavian Political Studies* **11**, 275 (1988).
- [L5] S. Grey, *Politics & Gender* **2**, 492 (2006).
- [L6] J. Xie, S. Sreenivasan, G. Korniss, W. Zhang, C. Lim, and B. K. Szymanski, *Phys. Rev. E* **84**, 011130 (2011).
- [L7] D. Mistry, Q. Zhang, N. Perra, and A. Baronchelli, *Phys. Rev. E* **92**, 042805 (2015).
- [L8] X. Niu, C. Doyle, G. Korniss, and B. K. Szymanski, *Scientific Reports* **7**, 41750 (2017).
- [L9] A. Baronchelli, *Royal Society Open Science* **5**, 172189 (2018).
- [L10] I. Iacopini, G. Petri, A. Barrat, and V. Latora, *Nature Communications* **10**, 1 (2019).
- [L11] B. Jhun, M. Jo, and B. Kahng, *Journal of Statistical Mechanics: Theory and Experiment* **2019**, 123207 (2019).
- [L12] G. F. de Arruda, G. Petri, and Y. Moreno, *Phys. Rev. Research* **2**, 023032 (2020).
- [L13] G. Ferraz de Arruda, M. Tizzani, and Y. Moreno, *Communications Physics* **4**, 24 (2021).
- [L14] A. Barrat, G. F. de Arruda, I. Iacopini, and Y. Moreno, “Social contagion on higher-order structures,” (2021), [arXiv:2103.03709](https://arxiv.org/abs/2103.03709) [physics.soc-ph].
- [L15] F. Battiston, G. Cencetti, I. Iacopini, V. Latora, M. Lucas, A. Patania, J.-G. Young, and G. Petri, *Physics Reports* **874**, 1 (2020), networks beyond pairwise interactions: Structure and dynamics.
- [L16] U. Alvarez-Rodriguez, F. Battiston, G. F. de Arruda, Y. Moreno, M. Perc, and V. Latora, *Nature Human Behaviour* **5**, 586 (2021).
- [L17] G. Ferraz de Arruda, M. Tizzani, and Y. Moreno, *Communications Physics* **4**, 24 (2021).
- [L18] D. J. Higham and H.-L. de Kergorlay, *Proceedings of the Royal Society A: Mathematical, Physical and Engineering Sciences* **477**, 20210232 (2021).
- [L19] D. J. Higham and H.-L. de Kergorlay, “Mean field analysis of hypergraph contagion model,” (2021), [arXiv:2108.05451](https://arxiv.org/abs/2108.05451) [math.DS].
- [L20] F. Battiston, E. Amico, A. Barrat, G. Bianconi, G. Ferraz de Arruda, B. Franceschiello, I. Iacopini, S. Kéfi, V. Latora, Y. Moreno, M. M. Murray, T. P. Peixoto, F. Vaccarino, and G. Petri, *Nature Physics* **17**, 1093 (2021).
- [L21] J.-H. Kim and K. I. Goh, “Higher-order components in hypergraphs,” (2022), [arXiv:2208.05718](https://arxiv.org/abs/2208.05718) [physics.soc-ph].
- [L22] R. Pastor-Satorras, C. Castellano, P. Van Mieghem, and A. Vespignani, *Rev. Mod. Phys.* **87**, 925 (2015).
- [L23] G. F. de Arruda, F. A. Rodrigues, and Y. Moreno, *Physics Reports* **756**, 1 (2018).
- [L24] M. Diani, *The Sociological Review* **40**, 1 (1992).
- [L25] R. Amato, L. Lacasa, A. Díaz-Guilera, and A. Baronchelli, *Proceedings of the National Academy of Sciences* **115**, 8260 (2018).
- [L26] F. A. Rodrigues, T. K. D. Peron, P. Ji, and J. Kurths, *Physics Reports* **610**, 1 (2016), the Kuramoto model in complex networks.
- [L27] P. S. Skardal, S. Adhikari, and J. G. Restrepo, “Multistability in coupled oscillator systems with higher-order interactions and community structure,” (2022), [arXiv:2207.00070](https://arxiv.org/abs/2207.00070) [nlin.AO].
- [L28] J. Poncela-Casasnovas, M. Gutiérrez-Roig, C. Gracia-Lázaro, J. Vicens, J. Gómez-Gardeñes, J. Perelló, Y. Moreno, J. Duch, and A. Sánchez, *Science Advances* **2**, e1600451 (2016).
- [L29] M. Gutiérrez-Roig, C. Gracia-Lázaro, J. Perelló, Y. Moreno, and A. Sánchez, *Nature Communications* **5**, 4362 (2014).
- [L30] D. M. O’Sullivan, E. O’Sullivan, M. O’Connor, D. Lyons, and J. McManus, *BMJ Innovations* **3**, 238 (2017).
- [L31] J. A. Caetano, G. Magno, M. Gonçalves, J. Almeida, H. T. Marques-Neto, and V. Almeida, in *Proceedings of the 10th ACM Conference on Web Science, WebSci ’19* (Association for Computing Machinery, New York, NY, USA, 2019) pp. 27–36.
- [L32] J. Rappaz, M. Catasta, R. West, and K. Aberer, in *ICWSM* (2018).
- [L33] M. Gladwell, *The Tipping Point: How Little Things Can Make a Big Difference* (Little, Brown and Company, 2000).
- [L34] D. Guilbeault and D. Centola, *Nature Communications* **12**, 4430 (2021).

REVIEWERS' COMMENTS

Reviewer #1 (Remarks to the Author):

The authors have revised the manuscript well according to my previous comment. Thus, I strongly recommend the manuscript to be published without further revision.

Reviewer #3 (Remarks to the Author):

The reviewers addressed all my points. Now the content is better organized, figures clearer, and the paper is overall improved with the new material and results. I believe it is ready for publication.